# LEARNING TO COLLABORATE

## ABSTRACT

In this paper, we focus on effective learning over a collaborative research network involving multiple clients. Each client has its own sample population which may not be shared with other clients due to privacy concerns. The goal is to learn a model for each client, which behaves better than the one learned from its own data, through secure collaborations with other clients in the network. Due to the discrepancies of the sample distributions across different clients, it is not necessarily that collaborating with everyone will lead to the best local models. We propose a learning to collaborate framework, where each client can choose to collaborate with certain members in the network to achieve a "collaboration equilibrium", where smaller collaboration coalitions are formed within the network so that each client can obtain the model with the best utility. We propose the concept of benefit graph which describes how each client can benefit from collaborating with other clients and develop a Pareto optimization approach to obtain it. Finally the collaboration coalitions can be derived from it based on graph operations. Our framework provides a new way of setting up collaborations in a research network. Experiments on both synthetic and real world data sets are provided to demonstrate the effectiveness of our method.

## 1 INTRODUCTION

Effective learning of machine learning models over a collaborative network of data clients has drawn considerable interest in recent years. Frequently, due to privacy concerns, we cannot simultaneously access the raw data residing on different clients. Therefore, distributed (Li et al., 2014) or federated learning (McMahan et al., 2017) strategies have been proposed, where typically model parameters are updated locally at each client with its own data and the parameter updates, such as gradients, are transmitted out and communicate with other clients. During this process, it is usually assumed that the participation in the network comes at no cost, i.e., every client is willing to participate in the collaboration. However, this is not always true in reality.

One example is the clinical research network (CRN) involving multiple hospitals (Fleurence et al., 2014). Each hospital has its own patient population. The patient data are sensitive and cannot be shared with other hospitals. If we want to build a risk prediction model with the patient data within this network in a privacy-preserving way, the expectation from each hospital is that a better model can be obtained through participating in the CRN compared to the one built from its own data collected from various clinical practice with big efforts. In this scenario, there has been a prior study showing that the model performance can decrease when collaborating with hospitals with very distinct patient populations due to negative transfer induced by sample distribution discrepancies (Wang et al., 2019; Pan & Yang, 2009).

With these considerations, in this paper, we propose a novel *learning to collaborate* framework. We allow the participating clients in a large collaborative network to form non-overlapping collaboration coalitions. Each coalition includes a subset of clients such that the collaboration among them can benefit their respective model performance. We aim at identifying the collaboration coalitions that can lead to a *collaboration equilibrium*, i.e., there are no other coalition settings that any of the individual clients can benefit more (i.e., achieve better model performance).

To obtain the coalitions that can lead to a collaboration equilibrium, we propose a Pareto optimization framework to identify the necessary collaborators for each client in the network to achieve its maximum utility. In particular, we optimize a local model associated with a specific client on the

(a) Benefit Graph    (b) Finding stable coalition    (c) Re-build Benefit Graph    (d) Collaboration Equilibrium

Figure 1: (a) The benefit graph on all clients; (b) Finding all stable coalitions and remove them; (c) reconstruct the benefit graph on the remaining clients; after $I^3$ is removed, $I^4$ re-identifies its necessary collaborators in $\{I^4, I^5, I^6\}$ which is $\{I^5\}$ as the added the red arrow from $I^5$ to $I^4$ in the figure; (d) iterate (b) and (c) until achieving collaboration equilibrium.

Pareto front of the learning objectives of all clients. Through the analysis of the geometric location of such an optimal model on the Pareto front, we can identify the necessary collaborators of each client. The relationships between each client and its necessary collaborators can be encoded in a *benefit graph*, in which each node denotes a client and the edge from $I^j$ to $I^i$ represents $I^j$ is one of the necessary collaborators for $I^i$, as exemplified in Figure 1 (a). Then we can derive the coalitions corresponding to the collaboration equilibrium through an iterative process introduced as follows. Specifically, we define a stable coalition as the minimum set such that its all involved clients can achieve its maximal utility. From the perspective of graph theory, these stable coalitions are actually the strongly connected components of the benefit graph. For example, $C = \{I^1, I^2, I^3\}$ in Figure 1 (b) is a stable coalition as all clients can achieve their best performance by collaborating with the clients in $C$ (compared with collaborating with other clients in the network). By removing the stable coalitions and re-building the benefit graph of the remaining client iteratively as shown in Figure 1 (b) and (c), we can identify all coalitions as in Figure 1 (d) and prove that the obtained coalitions can lead to a collaboration equilibrium.

We empirically evaluate our method on synthetic data, UCI adult (Kohavi, 1996), a classical FL benchmark data set CIFAR10 (Krizhevsky et al., 2009), and a real-world electronic health record (EHR) data repository eICU (Pollard et al., 2018), which includes patient EHR data in ICU from multiple hospitals. The results show our method significantly outperforms existing relevant methods. The experiments on eICU data demonstrate that our algorithm is able to derive a good collaboration strategy for the hospitals to collaborate.

## 2 RELATED WORK

### 2.1 FEDERATED LEARNING

Federated learning (FL) (McMahan et al., 2017) refers to the paradigm of learning from fragmented data without sacrificing privacy. In a typical FL setting, a global model is learned from the data residing in multiple distinct local clients. However, a single global model may lead to performance degradation on certain clients due to data heterogeneity. Personalized federated learning (PFL) (Kulkarni et al., 2020), which aims at learning a customized model for each client in the federation, has been proposed to tackle this challenge. For example, Zhang et al. (2020) proposes to adjust the weight of the objectives corresponding to all clients dynamically; Fallah et al. (2020) proposes a meta-learning based method for achieving an effectively shared initialization of all local models followed by a fine-tuning procedure; Shamsian et al. (2021) proposes to learn a central hypernetwork which can generate a set of customized models for each client. FL assumes all clients are willing to participate in the collaboration and existing methods have not considered whether the collaboration can really benefit each client or not. Without benefit, a local client could be reluctant to participate in the collaboration, which is a realistic scenario we investigate in this paper. One specific FL setup that is relevant to our work is clustered federated learning (Sattler et al., 2020; Mansour et al., 2020), which groups the clients with similar data distributions and trains a model for each client group. The scenario we are considering in this paper is to form collaboration coalitions based on the performance gain each client can get for its corresponding model, rather than sample distribution similarities.

### 2.2 MULTI-TASK LEARNING AND NEGATIVE TRANSFER

Multi-task learning (Caruana, 1997) (MTL) aims at learning shared knowledge across multiple inter-related tasks for mutual benefits. Typical examples include hard model parameter sharing (Kokkinos, 2017), soft parameter sharing (Lu et al., 2017), and neural architecture search (NAS) for a shared

model architecture (Real et al., 2019). However, sharing representations of model structures cannot guarantee model performance gain due to the existence of negative transfer, while we directly consider forming collaboration coalitions according to individual model performance benefits. In addition, MTL usually assumes the data from all tasks are accessible, while our goal is to learn a personalized model for each client through collaborating with other clients without directly accessing their raw data. It is worth mentioning that there are also clustered MTL approaches (Standley et al., 2020; Zamir et al., 2018) which assume the models for the tasks within the same group are similar to each other, while we want the clients within each coalition can benefit each other through collaboration when learning their respective models. Some works extend MTL to a multi-source domain adaptation (Wen et al., 2020; Duan et al., 2009) setting in which unlabeled data exist in some tasks. See detailed discussions in Appendix.

## 3 COLLABORATION LEARNING PROBLEM

We first introduce the necessary notations and definitions in Section 3.1, and then define the collaboration equilibrium we aim to achieve in Section 3.2.

### 3.1 DEFINITIONS AND NOTATIONS

Suppose there are $N$ clients $\boldsymbol{I} = \left\{ I^i \right\}_{i=1}^{N}$ in a collaborative network and each client is associated with a specific learning task $T^i$ based on its own data $D^i = \left\{ X^i, Y^i \right\}, i \in \{1, 2, ..., N\}$, where the input space $X^i$ and the output space $Y^i$ may or may not share across all $N$ clients. Each client pursues collaboration with others to learn a personalized model $M^i$ by maximizing its utility (i.e., model performance) without sacrificing data privacy. There is no guarantee that one client can always benefit from the collaboration with others, and the client would be reluctant to participate in the collaboration if there is no benefit. In the following, we describe this through a concrete example.

**No benefit, no collaboration.** Suppose the local data $\{D^i\}, i \in \{1, 2, ..., N\}$ owned by different clients satisfy the following conditions: 1) all local data are from the same distribution $D^i \sim \mathcal{P}$; 2) $D^1 \subset D^2 \subset D^3, ..., \subset D^N$. Since $D^N$ contains more data than other clients, $I^N$ cannot benefit from collaboration with any other clients, so $I^N$ will learn a local model using its own data. Once $I^N$ refuses to collaboration, $I^{N-1}$ will also work on its own as $I^{N-1}$ can only improve its utility by collaborating with $I^N$. $I^{N-2}$ will learn individually out of the same concerns. Finally, there is no collaboration among any clients.

Due to the discrepancies of the sample distributions across different clients, the best local model for a specific client is very likely to come from collaborating with a subset of clients rather than all of them. Suppose $U(I^i, C)$ denotes the model utility of client $I^i$ when collaborating with the clients in client set $C$. In the following, we define $U_{max}(I^i, C)$ as the maximum model utility that $I^i$ can achieve when collaborating with different subsets of $C$.

**Definition 1** (Maximum Achievable Utility (MAU)). *This is the maximum model utility for a specific client $I^i$ to collaborate with different subsets of client set $C$: $U_{max}(I^i, C) = \max_{C' \subset C} U(I^i, C')$.*

From Definition 1, MAU satisfies $U_{max}(I^i, C') \leq U_{max}(I^i, C)$ if $C'$ is a subset of $C$. Each client $I^i \in \boldsymbol{I}$ aims to identify its "optimal set" of collaborators from $\boldsymbol{I}$ to maximize its local utility, which is defined as follows.

**Definition 2** (Optimal Collaborator Set (OCS)). *A client set $C_{\boldsymbol{I}}^{opt}(I^i) \subset \boldsymbol{I}$ is an optimal collaborator set for $I^i$ if and only if $C_{\boldsymbol{I}}^{opt}(I^i)$ satisfies*

$$\forall C \subset \boldsymbol{I}, \ U(I^i, C_{\boldsymbol{I}}^{opt}(I^i)) \geq U(I^i, C); \tag{1a}$$

$$\forall C' \subsetneq C_{\boldsymbol{I}}^{opt}(I^i), \ U(I^i, C_{\boldsymbol{I}}^{opt}(I^i)) > U(I^i, C'). \tag{1b}$$

Eq.(1a) means that $I^i$ can achieve its maximal utility when collaborating with $C_{\boldsymbol{I}}^{opt}(I^i)$ and Eq.(1b) means that all clients in $C_{\boldsymbol{I}}^{opt}(I^i)$ are necessary. In this way, the relationships between any client $I^i \in \boldsymbol{I}$ and its optimal collaborator set $C_{\boldsymbol{I}}^{opt}(I^i)$ can be represented by a graph which is called the *benefit graph* (BG). Specifically, for a given client set $C \subset \boldsymbol{I}$, we use $BG(C)$ to denote its corresponding BG. For the example in Figure 1 (a), an arrow from $I^j$ to $I^i$ means $I^j \in C_{\boldsymbol{I}}^{opt}(I^i)$, e.g., $I^1 \rightarrow I^3$ means $I^1 \in C_{\boldsymbol{I}}^{opt}(I^3)$. For a client set $C$, if every member can achieve its maximum

model utility through the collaboration with other members within $C$ (without collaboration with other members outside $C$), then we call $C$ a *coalition*.

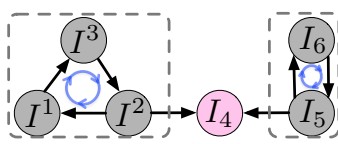

Figure 2: Forming coalitions for maximizing the local utility

**Forming coalitions for maximizing the local model utilities** Figure 2 shows an example BG with 6 clients. $I^3$ can achieve its optimal model utility by collaborating with $I^1$. Similarly, $I^1$ and $I^2$ can achieve their optimal model utility through collaborating with $I^2$ and $I^3$. In this case, $C = \{I^1, I^2, I^3\}$ denotes a collaboration coalition, and each client achieves its optimal utility by collaborating with other clients in $C$. If $I^1$ is taken out from $C$, $I^3$ will leave $C$ as well because it cannot gain any benefit through collaboration with others, and then $I^2$ will leave for the same reason. With this ring structure of $C$, none of the clients in $C$ can achieve its best performance without collaborating with the clients in $C$.

### 3.2 PROBLEM SETUP

As each client in $\boldsymbol{I}$ aims to maximize its local model utility by forming a collaboration coalition with others, all clients in $\boldsymbol{I}$ can form several non-overlapping collaboration coalitions. In order to derive those coalitions, we propose the concept of *collaboration equilibrium* (CE) as follows.

Suppose we have a set of coalitions $S = \{C^0, C^1, ...C^K\}$ such that $\bigcup_{k=1}^{K} C^k = \boldsymbol{I}$ and $C^{k_1} \bigcap C^{k_2} = \emptyset$ for $\forall k_1 \neq k_2$, then we say $S$ reaches CE if it satisfies the following two axioms.

**Axiom 1** (Inner Agreement). *All collaboration coalitions satisfy inner agreement, i.e.,*

$$\forall C \in S, \ \forall C' \subsetneq C, \ \exists I^i \in C', \ s.t., \ U_{max}(I^i, C') < U_{max}(I^i, C) \tag{2}$$

From Axiom 1, inner agreement emphasizes that the clients of each coalition agree to form this coalition. It gives the necessary condition for a collaboration coalition to be formed such that any of the subset $C' \subsetneq C$ can benefit from the collaboration with $C \backslash C'$. Eq.(2) tells us that there always exists a client in $C'$ that opposes leaving $C$ because its utility will go down if $C'$ is split from $C$. In this way, inner agreement guarantees that all coalitions will not fall apart or the clients involved will suffer. For example, $S = \{\{I^1, I^2, I^3, I^4\}, \{I^5, I^6\}\}$ in Figure 2 does not satisfy inner agreement, because the clients in the subset $C' = \{I^1, I^2, I^3\}$ achieves their optimal utility in $C'$ and can leave $C = \{I^1, I^2, I^3, I^4\}$ without any loss.

**Axiom 2** (Outer Agreement). *The collaboration strategy should satisfy outer agreement, i.e.,*

$$\forall C' \notin S, \ \exists I^i \in C', \ s.t. \ U_{max}(I^i, C') \leq U_{max}(I^i, C) \ (I^i \in C \in S) \tag{3}$$

From Axiom 2, outer agreement guarantees that there is no other coalition $C'$ which can benefit each client involved more than $S$ achieves. Eq.(3) tells us that if $C'$ is a coalition not from $S$, there always exists a client $I^i$ and a coalition in $C \in S$ such that $I^i$ can benefit more. The collaboration strategy $S = \{\{I^1, I^2, I^3\}, \{I^4\}, \{I^5, I^6\}\}$ in Figure 2 is a CE in which the clients in $\{I^1, I^2, I^3\}$ and $\{I^5, I^6\}$ achieve their optimal model utility. Though $I^4$ does not achieve its maximum model utility in $\{I^4\}$, there is no other coalitions which can attract $I^2$ and $I^5$ to form a new coalition with $I^4$. Therefore, all clients have no better choice but agree upon this collaboration strategy.

Our goal is to obtain a collaboration strategy to achieve CE which satisfies Axiom 1 and Axiom 2, so that all clients achieve their optimal model utilities in the collaboration coalition. For the nature of CE, we also share our extended discussions in Appendix. In the next section, we introduce our algorithm in detail on 1) how to derive a collaboration strategy that can achieve CE from the benefit graph and 2) how to construct the benefit graph.

## 4 COLLABORATION EQUILIBRIUM

In this section, we will introduce our framework on learning to collaborate. Firstly, we propose an iterative graph-theory-based method to achieve CE based on a given benefit graph.

### 4.1 Achieving Collaboration Equilibrium Given the Benefit Graph

In theory, there are $B_N$ collaboration strategies for partitioning $N$ clients into several coalitions, where $B_N$ is the Bell number which denotes how many solutions for partitioning a set with $N$ elements (Bell, 1934). While exhaustive trying all partitions has exponential time complexity, in this section, we propose an iterative method for deriving a collaboration strategy that achieves CE with polynomial time complexity. Specifically, at each iteration, we search for a *stable coalition* which is formally defined in Definition 3 below, then we remove the clients in the *stable coalition* and re-build the benefit graph for the remaining clients. The iterations will continue until all clients are able to identify their own coalitions.

**Definition 3** (Stable Coalition). *Given a client set $I$, a coalition $C^s$ is stable if it satisfies*

1. *Each client in $C^s$ achieves its maximal model utility, i.e.,*
$$U_{max}(I^i, C^s) = U_{max}(I^i, I) \, \forall I^i \in C^s. \tag{4}$$

2. *Any sub coalition $C' \subset C^s$ cannot achieve the maximal utility for all clients in $C'$, i.e.,*
$$\forall C' \subsetneqq C^s, \, \exists I^i \in C', \, s.t., \, U_{max}(I^i, C') < U_{max}(I^i, C^s). \tag{5}$$

From Definition 3, Eq.(4) means that any client in a stable coalition cannot improve its model utility further. Eq.(5) states that this coalition is *stable* as any sub coalition $C'$ can benefit from $C^s \backslash C'$. Therefore any sub coalition has no motivation to leave $C^s$. Eq.(4) implies that a stable coalition will not welcome any other clients to join as others will not benefit the clients in $C^s$ further. In Figure 2, $\{I^1, I^2, I^3\}$ and $\{I^5, I^6\}$ are the two stable coalitions. In order to identify the stable coalitions from the benefit graph, we first introduce the concept of strongly connected component in a directed graph.

**Definition 4** (Strongly Connected Component (Tarjan, 1972)). *A subgraph $G'$ is a strongly connected component of a given directed graph $G$ if it satisfies: 1) It is strongly connected, which means that there is a path in each direction between each pair of vertices in $G'$; 2) It is maximal, which means no additional vertices from $G$ can be included in $G'$ without breaking the property of being strongly connected.*

Then we derive a graph-based method to obtain the collaboration coalitions that can achieve collaboration equilibrium by identifying all stable coalitions iteratively according to Theorem 1 below.

**Theorem 1.** *(Proof in Appendix) Given a client set $I$ and its $BG(I)$, the stable coalitions are strongly connected components of $BG(I)$.*

With Theorem 1, we need to identify all strongly connected components of $BG(I)$, which can be achieved using the **Tarjan** algorithm (Tarjan, 1972) with time complexity $O(V + E)$, where $V$ is the number of nodes and $E$ is the number of edges. Then following Eq.(4), we judge whether a strongly connected component is a stable coalition by checking whether all clients have achieved their maximal model utility. A stable coalition $C^s$ has no interest to collaborate with other clients, so $C^s$ will be removed and the remaining clients $I \backslash C^s$ will continue to seek collaborations until all $N$ clients find their coalitions. In this way, we can achieve a partitioning strategy, with the details shown in Algorithm 1 in the Appendix.

**Theorem 2.** *(Proof in Appendix) The collaboration strategy obtained above achieves collaboration equilibrium.*

The clients in all stable coalitions found in each iteration cannot improve their model utility further and will not collaborate with others because there are no additional benefits. Therefore, the collaboration strategy is approved by all clients. The iterative method achieves CE considering the $BG$ varies in each iteration after removing the stable coalitions, which can be time-consuming because we need to redefine $BG$ in each iteration by re-learning an optimal personalized model for each remaining client.

**Assumption 1.** *The benefit graph of a subset $C \subset I$ ($BG(C)$) is the subgraph of the $BG(I)$.*

Assumption 1 claims that the benefit graph of the remaining clients $I \backslash C^s$ keeps unchanged when the subgraph $BG(C^s)$ is split from $BG(I)$. It implies that for each pair of clients $I^i$ and $I^j$, whether $I^i$ is one of the optimal collaborators for $I^j$ will not be affected by other clients. In this case, we do not need to re-build the benefit graph and have the following corollary.

**Corollary 1.** *(proof in Appendix) When Assumption 1 holds, the strongly connected components of $BG(I)$ leads to a collaboration equilibrium.*

## 4.2 Determine the Benefit Graph by Specific Pareto Optimization

The benefit graph of $N$ clients consists of the clients and their corresponding OCS. However, each client has $2^{N-1}$ collaborator sets and it's hard to determine which one is the OCS. Exhaustive trying all sets to determine the OCS may be impractical, especially when there are many clients. To identify the OCS effectively and efficiently, we propose Specific Pareto Optimization (SPO) to alternately perform the following two steps: 1). learn a Pareto model (defined below) given the weight of all clients $d$; 2). optimize the weight vector $d$ to search for a best model $M^*(I^i)$ by gradient descent.

**Definition 5** (Pareto Solution and Pareto Front). *We consider $n$ objectives corresponding to $n$ clients: $l_i : \mathbb{R}^n \rightarrow \mathbb{R}_+, i = \{1, 2, ..., n\}$. Given a learned hypothesis $h$, suppose the loss vector $l(h) = [l_1, l_2, ..., l_n]$ represents the utility loss on $n$ clients with hypothesis $h \in \mathcal{H}$, we say $h$ is a Pareto Solution if there is no hypothesis $h'$ that dominates $h$, often called Pareto optimality, i.e.,*

$$\nexists h' \in \mathcal{H}, \; s.t. \; \forall i : l_i(h') \leq l_i(h) \; and \; \exists j : l_j(h') < l_j(h).$$

*In a collaboration network with $N$ clients $\{I^i\}_{i=1}^N$, as each client has its own learning task which can be formulated as a specific objective, we use $P(\{I^1, I^2, ..., I^N\})$ to represent the Pareto Front (PF) of the client set $\{I^i\}_{i=1}^N$ formed by all Pareto hypothesis.*

**Learning a best model and identify the OCS by SPO**  To search for an optimal model $M^*(I^i)$ for $I^i$, we propose to firstly learn the empirical Pareto Front by a hypernetwork denoted as $HN$ [1]. As each client owns its objective, given the weight of all clients (objectives), the learned PT by $HN$ outputs a Pareto model $M$,

$$M = HN(d), \tag{6}$$

where $d = [d^1, d^2, ..., d^N]$ satisfying $\sum_{i=1}^N d^i = 1$ and $d^i$ denotes the weight of the objective $l_i$. Each $d$ corresponds to a specific Pareto model $M$, and all Pareto models $M$ satisfy Pareto optimality that the losses on the training data of all clients cannot be further optimized.

Though the Pareto models maximize the utilization of the training data from all clients, they may not be the best models on the true (test) data distribution. For example, $HN(d_4)$ achieves a minimum loss on the training data of $I^1$ shown in Figure 3 (b), but it is not the optimal model on the true data distribution of $I^1$ shown in Figure 3 (c). Therefore, we propose to search for a best Pareto model that achieves the best performance on the true (validation) data set of the target client $I^i$, i.e.,

$$M^*(I^i) = HN(d^*), \quad \text{where } d^* = \arg\max_d \text{Per}(HN(d), I^i), \tag{7}$$

where $\text{Per}(HN(d), I^i)$ denotes the performance of $HN(d)$ on the validation data set of $I^i$. We identify the OCS of $I^i$ based on the optimized weight of all clients $d^*$. For example shown in Figure 3(c), the best model is $h_1^*$ for client $I^1$ and the corresponding optimized weight $d^* = d_3 = [0.7, 0.3, 0]$. So the OCS of $I^1$ consists of the clients with non-zero weights, which is $\{I^1, I^2\}$.

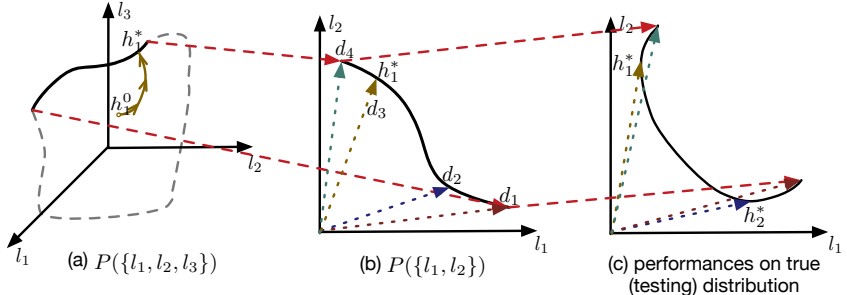

Figure 3: (a) the loss plane of $P(\{l_1, l_2, l_3\})$ learned from training data of $\{I^1, I^2, I^3\}$; (b) the loss curve of $P(\{l_1, l_2\})$ learned from training data of $\{I^1, I^2\}$ is embedded in the loss plane of $P(\{l_1, l_2, l_3\})$; $d_1, d_2, ., d_4$ are 4 vectors corresponding to 4 Pareto models in $P(\{l_1, l_2\})$; (3) the performances of the models $h \in P(\{l_1, l_2\})$ on true (testing) distributions and $h_1^*$ ($h_2^*$) achieves the optimal performance on client $I^1$ ($I^2$) corresponding to $d_3$ ($d_2$) in (b).

---

[1]More information please refer to (Navon et al., 2020)

**Proposition 1** (Pareto Front Embedding Property). *(proof in Appendix) Suppose* $\boldsymbol{l'^*} = [l_i(h'^*)], i \in C'$ *and* $\boldsymbol{l^*} = [l_i(h^*)], i \in C$ *are the loss vectors achieved by the PFs* $P(C')$ *and* $P(C)$ *where* $C' \subset C$, *then*

$$\forall h'^* \in P(C'), \ \exists h^* \in P(C), \ s.t., \ l_i(h'^*) = l_i(h^*) \ \forall i \in C'. \tag{8}$$

From Proposition 1, the loss vectors achieved by the PF of a sub-coalition are embedded in the loss vectors of the full coalition, such as the loss curve of $P(\{l_1, l_2\})$ is in the loss plane of $P(\{l_1, l_2, l_3\})$ shown in Figure 3 (a) and (b).

**Explanation of SPO from a geometric point of view** Intuitively, if the best model $M^*(I^i)$ searched on the PF of all clients $P(\boldsymbol{I})$ belongs to the PF of a sub coalition $P(C)$ simultaneously, i.e.,

$$M^*(I^i) \in P(C) \quad \text{and} \quad M^*(I^i) \in P(I), \tag{9}$$

then the clients $\boldsymbol{I} \backslash C$ are not necessary for obtaining $M^*(I^i)$. However, exhaustively trying the PF of all sub-coalitions can have exponential time complexity. From Proposition 1, the PF of full coalition $\boldsymbol{I}$ contains the PF of all sub-coalitions. Therefore, we propose SPO to search for a best model $M^*(I^i)$ on the PF of full coalition $\boldsymbol{I}$, and identify whether $M^*(I^i)$ belongs to the PF of a sub-coalition using the optimized weight vector $\boldsymbol{d}$. For example, suppose there are three clients and the three corresponding objectives achieved by the PF are shown in Figure 3(a). The model $h_1^*$ in Figure 3(a) is on the PF $P(\{l_1, l_2, l_3\})$. Since the corresponding weight $\boldsymbol{d} = [0.7, 0.3, 0]$, from Figure 3(b), $h_1^*$ is also on the PF $P(\{l_1, l_2\})$, so $I^3$ is not a necessary client and the OCS of $I^1$ is $\{l_1, l_2\}$. More discussion about optimality and implementation details are in Appendix.

## 5 Experiments

To intuitively demonstrate the motivation of *collaboration equilibrium* and the effectiveness of SPO, we conduct experiments on synthetic data, a real-world UCI dataset **Adult** (Kohavi, 1996) and a benchmark data set **CIFAR10** (LeCun et al., 1998). Moreover, we verify the practicability of our framework on a real-world multiple hospitals collaboration network using the electronic health record (EHR) data set **eICU** (Pollard et al., 2018). As SPO aims to achieve an optimal model utility by optimizing the personalized model on the PF of all clients, we use SPO to denote the model utility achieved by SPO. According to the OCS determined by SPO we achieve a CE for all clients and the model utility of each client in the CE can be different from the utility achieved by SPO. We use CE to denote the model utility achieved in the CE without causing further confusion. To verify the practicability of our framework, we also compare the time consumption and provide more experimental results in Appendix.

### 5.1 Synthetic Experiments

**Synthetic data** Suppose there are 6 clients in the collaboration network. The synthetic features owned by each client $I^i$ are generated by $\mathbf{x} \sim \mathcal{U}[-1.0, 1.0]$; the ground-truth weights $\mathbf{u}_i = \mathbf{v} + \mathbf{r}_i$ are samples as $\mathbf{v} \sim \mathcal{U}[0.0, 1.0], \mathbf{r}_i \sim \mathcal{N}(\mathbf{0}_d, \rho^2)$ where $\rho^2$ represents the client variance (if $\rho^2$ increases, the data distribution discrepancy among clients will increase). Labels of the clients are observed with i.i.d noise $\epsilon \sim \mathcal{N}(0, \sigma^2)$. To generate conflicting learning tasks assigned to different clients, we flip over the label of some clients: $y = \mathbf{u}_i^\top \mathbf{x} + \epsilon, i \in \{0, 1, 2\}$ and $y = -\mathbf{u}_i^\top \mathbf{x} + \epsilon, i \in \{3, 4, 5\}$.

From Table 1, when there are fewer samples ($n = 2000$) and less distribution discrepancy $\rho = 0.1$ in the client set $\{I^0, I^1, I^2\}$ or $\{I^3, I^4, I^5\}$ with similar label generation process, these clients collaborate with others to achieve a low MSE. In this case, the OCS of each client is the clients with similar learning tasks and we achieve CE as $S = \{\{I^i\}_{i=0}^2, \{I^i\}_{i=3}^5\}$ as shown in the top of Figure 4 (a). With the increase of the number of samples and the distribution discrepancy, collaboration cannot benefit the clients and all clients will learn individually on their own data. Therefore, when $n = 20000$ and $\rho = 1.0$, the OCS of each client is itself and the collaboration strategy $S = \{\{I^0\}, \{I^1\}, ., \{I^5\}\}$ leads to a CE as shown in the bottom of Figure 4 (a).

**UCI adult data** *adult* contains more than 40000 adult records and the task is to predict whether an individual earns more than 50K/year given other features (e.g., age, gender, education, etc.). Following the setting in (Li et al., 2019; Mohri et al., 2019), we split the data set into two clients. One is PhD client ($I^1$) in which all individuals are PhDs and the other is non-PhD client ($I^0$). In this

experiment, we implement SPO on this data set and compare the performance with existing relevant methods AFL (Mohri et al., 2019) and q-FFL (Li et al., 2019)[2].

The two clients $I^0$ and $I^1$ have different data distribution and non-PhD client has more than 30000 samples while PhD client has about 500 samples. From Table 2, SPO achieves higher accuracy compared to baselines especially on PhD clients (77.0). non-PhD client achieves an optimal accuracy (83.5) by local training. Therefore, PhD client improves its performance by collaborating with non-PhD client while the performance of non-PhD client declines. The benefit graph is shown in the top of Figure 4 (b). The CE is non-collaboration as in the bottom of Figure 4 (b) and the model of both clients in the CE are trained individually.

## 5.2 BENCHMARK EXPERIMENTS

We compare our method with previous personalized federated learning (PFL) methods on CIFAR-10 (Krizhevsky et al., 2009)[3]. Following the setting in (McMahan et al., 2016), we simulate non-i.i.d environment by randomly assigning two classes to each client among ten total classes. Baselines we evaluate are as follows: (1) Local training on each client; (2) FedAvg (McMahan et al., 2016); (3) Per-FedAvg (Fallah et al., 2020), a meta-learning based PFL algorithm. (4) pFedMe (T Dinh et al., 2020), a PFL approach which adds a Moreau-envelopes loss term; (5) LG-FedAvg (Liang et al., 2020) PFL method with local feature extractor and global output layers; (6) FedPer (Arivazhagan et al., 2019), a PFL approach that learns personal classifier on top of a shared feature extractor; (7) pFedHN (Shamsian et al., 2021), a PFL approach that generates models by training a hyper-network. In all experiments, our target network shares the same architecture as the baseline models. For each client, we split 87% of the training data for learning a Pareto Front by collaborating with the others and the remaining 13% of the training data for optimizing the vector $d$ to reach an optimal model as shown in Figure 3 (c). More implementation details are in Appendix.

Table 3 reports the results of all methods. FedAve achieves a lower accuracy (51.4) compared to local training (86.46) which means that training a global model can hurt the performance of each client. Compared to other PFL methods in Table 3, SPO reaches an optimal model on the PF of all objectives and achieves a higher accuracy (92.47). As the features learned from the images are transferable though there is a label shift among all clients, the collaboration among all clients leads to a more efficient feature extractor for each client. Therefore, the benefit graph of this collaboration network is a fully connected graph and the collaboration equilibrium is that all clients form a full coalition for collaboration as shown in Figure 4 (c). In this experiment, the accuracy model of each clients in CE equals to the accuracy achieved by SPO.

Table 1: Synthetic

| I | $n = 2000, \rho = 0.1$ | | $n = 20000, \rho = 1.0$ | |
|---|---|---|---|---|
| | OCS | CE (MSE) | OCS | CE (MSE) |
| $I^0$ | $\{I^0, I^1, I^2\}$ | 0.24±0.08 | $\{I^0\}$ | 1e-4±.0 |
| $I^1$ | $\{I^0, I^1, I^2\}$ | 0.26±0.08 | $\{I^1\}$ | 1e-4±.0 |
| $I^2$ | $\{I^0, I^1, I^2\}$ | 0.24±0.04 | $\{I^2\}$ | 1e-4±.0 |
| $I^3$ | $\{I^3, I^4, I^5\}$ | 0.26±0.07 | $\{I^3\}$ | 1e-4±.0 |
| $I^4$ | $\{I^3, I^4, I^5\}$ | 0.26±0.09 | $\{I^4\}$ | 1e-4±.0 |
| $I^5$ | $\{I^3, I^4, I^5\}$ | 0.26±0.03 | $\{I^5\}$ | 1e-4±.0 |

Table 2: Adult

| methods | Accuracy | |
|---|---|---|
| | $I^0$ | $I^1$ |
| AFL | 82.6 ± .5 | 73.0 ± 2.2 |
| q-FFL | 82.4 ± .1 | 74.4 ± .9 |
| local | **83.5 ± .0** | 66.9 ± 1.0 |
| SPO (ours) | 82.8 ± .3 | **77.0 ± .7** |
| CE (ours) | **83.5 ± .0** | 66.9 ± 1.0 |

Table 3: CIFAR10

| methods | accuracy |
|---|---|
| Local | 86.46 ± 4.02 |
| FedAve | 51.42 ± 2.41 |
| Per-FedAve | 76.65 ± 4.84 |
| FedPer | 87.27 ± 1.39 |
| pFedMe | 87.69 ± 1.93 |
| LG-FedAve | 89.11 ± 2.66 |
| pFedHN | 90.83 ± 1.56 |
| SPO (ours) | **92.47 ± 4.80** |
| CE (ours) | **92.47 ± 4.80** |

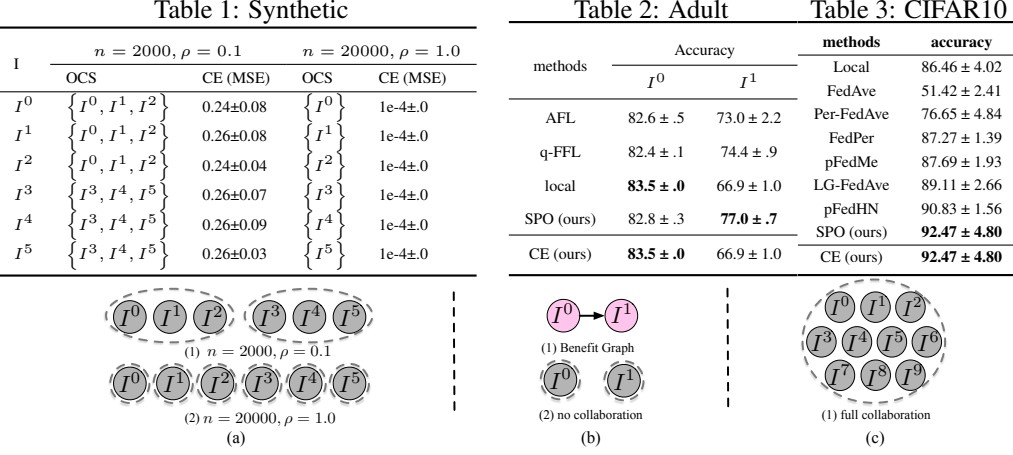

Figure 4: Collaboration equilibrium on synthetic data, Adult and CIFAR10.

## 5.3 HOSPITAL COLLABORATION

eICU (Pollard et al., 2018) is a clinical data set collecting the patients about their admissions to ICUs with hospital information. Each instance is a specific ICU stay. We follow the data pre-processing

---

[2]The results of baselines are from (Li et al., 2019)

[3]The results of baselines are from (Shamsian et al., 2021)

procedure in (Sheikhalishahi et al., 2019) and naturally treat different hospitals as local clients. We conduct the task of predicting in-hospital mortality which is defined as the patient's outcome at the hospital discharge. This is a binary classification task, where each data sample spans a 1-hour window. In this experiment, we select 5 hospitals with more patient samples (about 1000) $\left\{I^i\right\}_{i=0}^4$ and 5 hospitals with less patient samples $\left\{I^i\right\}_{i=5}^9$. Due to label imbalance (more than 90% samples have negative labels), we use AUC to measure the utility for each client as in (Sheikhalishahi et al., 2019). For all methods, we use the ANN as the network structure as in (Sheikhalishahi et al., 2019).

Table 4: eICU

| methods | AUC | | | | | | | | | |
|---|---|---|---|---|---|---|---|---|---|---|
| | $I^0$ | $I^1$ | $I^2$ | $I^3$ | $I^4$ | $I^5$ | $I^6$ | $I^7$ | $I^8$ | $I^9$ |
| Local | 66.89 | 85.03 | 61.83 | 68.83 | 82.31 | 59.65 | 67.78 | 40.00 | 61.90 | 70.00 |
| FedAve | 71.92 | 89.36 | **81.00** | **73.89** | 80.23 | 70.18 | 52.22 | 40.00 | 61.90 | 75.00 |
| SPO (ours) | **76.35** | **91.80** | 80.28 | 70.52 | **86.93** | **82.46** | **71.11** | **40.00** | **76.19** | **83.33** |
| CE (ours) | 77.93 | 87.28 | 70.47 | 70.64 | 83.48 | 64.92 | 68.89 | 45.00 | 61.90 | 70.00 |

The model AUC of each hospital is reported in Table 4. Because of the lack of patient data for each hospital, Local achieves a relatively lower AUC compared to FedAve and SPO. While patient populations vary substantially from hospital to hospital, SPO learns a personalized model for each hospital and outperforms FedAve from Table 4.

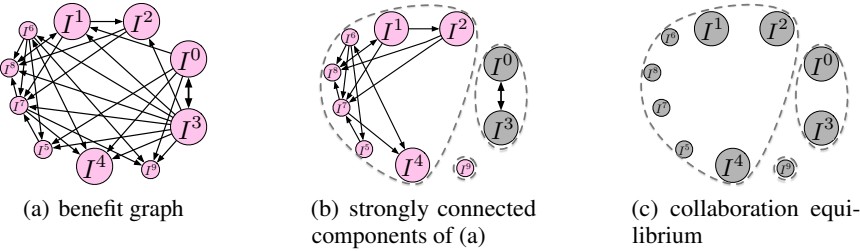

(a) benefit graph    (b) strongly connected components of (a)    (c) collaboration equilibrium

Figure 5: Collaboration Equilibrium of 10 real hospitals

**Collaboration Equilibrium**    The benefit graph of all hospitals determined by SPO and its corresponding strongly connected components are shown in Figure 5(a) and 5(b). From Figure 5(a), $I^0$ and $I^3$ are the unique necessary collaborator for each other, $C^1 = \left\{I^0, I^3\right\}$ is the first identified stable coalition as shown in Figure 5(b); $I^9$ is a tiny clinic that cannot contribute to any hospitals, so no hospital is willing to collaborate with it and $I^9$ learns a local model with its own data by forming a simple coalition $C^2 = \left\{I^9\right\}$. The benefit graph of the remaining clients are shown in Figure 5(b). On the one hand they cannot benefit $I^3$ or $I^0$ so they cannot form coalitions with them, on the other hand they refuse to contribute $I^9$ without any charge. They choose form the coalition $C^3 = \left\{I^1, I^2, I^4, I^5, I^6, I^7, I^8\right\}$ to maximize their AUC. Therefore, the CE in this hospital collaboration network is achieved by the collaboration strategy $S = \{C^1, C^2, C^3\}$ and the model AUC of each client in the CE is in Table 4.

Comparing Figure 5(b) and 5(c), the strongly connected components of the benefit graph leads to a CE. This empirically verifies that Assumption 1 is reasonable. Meanwhile, CE guarantees that every client in its coalition will not collaborate with harmful clients, so the client may achieve a higher utility in a CE compared to collaborating with everyone such as the AUC of $I^0$ in CE (77.93) is higher than in SPO (76.35).

## 6 CONCLUSION

In this paper, we propose a *learning to collaborate* framework to achieve collaboration equilibrium such that any of the individual clients cannot improve their performance further. Comprehensive experiments on benchmark and real-world data sets demonstrated the validity of our proposed framework. In our study, some small clients could be isolated as they cannot benefit others. Our framework can quantify both the benefit to and the contribution from each client in a network. In practice, such information can be utilized to either provide incentives or to impose charges on each client, to facilitate and enhance the foundation of the network or coalition.

ETHICS STATEMENT

We propose a novel collaboration framework for multiple clients, which is based the principle that each member in the coalition benefits others while aims to maximize its utility. We have read the codes of ethics and make sure this paper conforms to them. The procedures of data processing and experiments comply with requirements. Moreover, we also share our extended discussions about the applications and limitations of our method in Appendix.

REPRODUCIBILITY STATEMENT

Our all instructions and training details needed to reproduce the experimental results are in Appendix. The links of public data sets are also provided. We report the resources used in Appendix. The source code of our method will be made public when it is published. For all theoretical results, we also give rigorous analysis, detailed proofs, and comprehensive discussions in Appendix.

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

## A  PROOFS OF ALL THEORETICAL RESULTS

### A.1  PROOF OF THEOREM 1

To prove Theorem 1, we first prove that the benefit graph of a stable coalition is strongly connected shown in Lemma 1.

**Lemma 1.** *For a given client set $I$, the benefit graph of each stable coalition $C^s$ is strongly connected, which means that there is a path in each direction between each pair of vertices in $BG(C^s)$.*

*Proof.* We will prove the Lemma 1 by contradiction. Given a stable coalition $C^s$, suppose there exsits a pair of vertices $I^1, I^2 \in BG(C^s)$ such that there is no path from $I^1$ to $I^2$, which is denoted as $\nexists p$, s.t., $I^1 \to I^2$ for expressive clearly. We split $C^s$ into two sub-coalitions $C'$ and $C^s \backslash C'$ depending on whether the clients have paths to $I^2$. The clients in $C'$ have no path to $I^2$, and the clients in $C^s \backslash C'$ have paths to $I^2$.

$$
\begin{aligned}
&\forall I^i \in C', \nexists p, \text{ s.t., } I^i \to I^2 \\
&\forall I^i \in C^s \backslash C', \exists p, \text{ s.t., } I^i \to I^2.
\end{aligned}
\tag{10}
$$

Obviously, $C'$ and $C^s \backslash C'$ are not empty because $I^1 \in C'$ and and $I^2 \in C^s \backslash C'$. From Eq.(5) in the main text in the Definition of *stable coalition*, each client $I^i \in C^s$ achieves its maximal utility by collaborating with others in $C^s$ which means that

$$
\forall I^i \in C^s, \ \forall I^j \in C_I^{opt}(I^i), \ I^j \in C^s.
\tag{11}
$$

From the Definition of *Optimal Collaborator Set* (OCS), each client $I^i$ in the OCS of $I^2$ has an edge from $I^i$ to $I^2$,

$$
\forall I^i \in C_I^{opt}(I^2), \ \exists p, \text{ s.t., } I^i \to I^2,
\tag{12}
$$

Therefore, because each client $I^i \in C_I^{opt}(I^2)$ is in $C^s \backslash C'$, $I^2$ can achieve its maximal utility in $C^s \backslash C'$. Morever, we can prove that $I^i \in C_I^{opt}(I^2)$ achieves its maximal utiltiy in $C^s \backslash C'$ because each client $I^i \in C_I^{opt}(I^2)$ and its corresponding OCS are in $C^s \backslash C'$,

$$
\forall I^i \in C_I^{opt}(I^2), \ \forall I^j \in C_I^{opt}(I^i), \ \exists p, \ I^j \to I^i,
\tag{13a}
$$

$$
\forall I^i \in C_I^{opt}(I^2), \ \exists p, \text{ s.t., } I^i \to I^2,
\tag{13b}
$$

$$
\Rightarrow \ \forall I^i \in C_I^{opt}(I^2), \ \forall I^j \in C_I^{opt}(I^i), \ \exists p, \ I^j \to I^i \to I^2
\tag{13c}
$$

From Eq.(13), the clients in the OCS of $I^i$ ($I^i \in C_I^{opt}(I^2)$) are in $C^s \backslash C'$ and $I^i$ can achieve its optimal utility. Based on the same analysis, for each client $I^i \in C^s \backslash C'$, its OCS $C_I^{opt}(I^i)$ is in $C^s \backslash C'$ because all clients in $C_I^{opt}(I^i)$ have a path to $I^2$. So we have the conclusion that all clients in $C^s \backslash C'$ achieves its optimal utility by collaborating with others in $C^s \backslash C'$. However, this contradicts Eq.(6) in the main text in the Definition of *stable coalition*. Therefore, we prove that the benefit graph $BG(C^s)$ of the stable coalition $C^s$ is strongly connected as in Lemma 1. $\qquad\square$

From Lemma 1, the $BG(C^s)$ is strongly connected, then we will prove that $BG(C^s)$ is a strongly connected component of $BG(\boldsymbol{I})$ by pointing out that $BG(C^s)$ is maximal which means that no additional vertices from $\boldsymbol{I}$ can be included in $BG(C^s)$ without breaking the property of being strongly connected.

*Proof.* We will prove that $BG(C^s)$ is maximal by contradiction. Suppose there is another client $I^0 \in \boldsymbol{I} \backslash C^s$ which can be added in $BG(C^s)$ without breaking the property of being strongly connected. Therefore, there exists $I^i \in C^s$ which has an edge from $I^0$ to $I^i$. This means that $I^0$ is one of the necessary collaborators of $I^i$ ($I^0 \in C_I^{opt}(I^i)$) and $I^0$ cannot achieve its optimal utility in $C^s$ without collaborating with $I^0$. However, this contradicts Eq.(5) in the main text in the Definition of *stable coalition* that all clients in $C^s$ can achieve its optimal utility by collaborating with others in $C^s$. Therefore, we prove that $BG(C^s)$ is a strongly connected component as stated in Theorem 1. $\qquad\square$

## A.2 Proof of Theorem 2

*Proof.* **1. All coalitions in $S = \{C^0, C^1, ..., C^k\}$ satisfies Inner Agreement as in Axiom 1.**

As each coalition $C^i \in S$ is a stable coalition, from Eq.(6) of the Definition of *stable coalition*, we have

$$\forall C^i \in S, \ \forall C' \subsetneqq C^i, \ \exists I^i \in C', \ \text{s.t.,} \ U_{max}(I^i, C') < U_{max}(I^i, C^i). \tag{14}$$

Then Eq.(14) is the definition of *Inner Agreement* in Eq.(3) in the main text. Therefore, we prove that all collaboration coalitions in $S$ satisfy inner agreement.

**2. The collaboration strategy $S$ satisfies Outer Agreement as in Axiom 2.**

We will prove it by contradiction. Suppose there exists a new nonempty coalition $C' = \{I^{i^0}, I^{i^1}, ..., I^{i^k}\} \notin S$ which satisfies

$$\forall I^i \in C', \ U_{max}(I^i, C') > U_{max}(I^i, C^j) \ (I^i \in C^j \in S). \tag{15}$$

Suppose that $C^0$ denotes the coalition in the first iteration. Then $\forall I^i \in C^0$, $I^i$ achieves its maximal utility in the client set $\boldsymbol{I}$ and cannot increase its model utility further. So we have

$$\forall I^i \in C', \ I^i \notin C^0. \tag{16}$$

The clients in the coalition $C^0 \in S$ identified in the first iteration have no interest in collaborating with others and will be removed. Suppose $C^1$ denotes the coalition in the first iteration. $\forall I^i \in C^1$, $I^i$ achieves its maximal utility in the client set $\boldsymbol{I} \backslash C^0$ and cannot increase its model utility further. So we have

$$\forall I^i \in C', \ I^i \notin C^1. \tag{17}$$

Then $C^1$ will be removed and we have $\forall I^i \in C', \ I^i \notin I^2$.. Based on the same analysis, we finally have

$$\begin{aligned} &\forall I^i \in C', \ \forall C^j \in S, \ I^i \notin C^j, \\ &\Rightarrow C' = \emptyset. \end{aligned} \tag{18}$$

Therefore, we prove that there is no nonempty $C'$ satisfying Eq.(15) and $S$ satisfies Outer Agreement. $\qquad\square$

## A.3 Proof of Corollary 1

Firstly, we prove that all strongly connected components are disjoint as in Lemma 2.

**Lemma 2.** *Given a directed graph $G$ and all its strongly connected components $\{C^0, ...C^k\}$, each pair of its strongly connected components $C^i$ and $C^j$ satisfies $C^i \cap C^j = \emptyset$.*

*Proof.* We will prove it by contradiction. Suppose there exists a pair of the strongly connected components which satisfies $I^0 \in (C^i \cap C^j)$ and $C = C^i \cup C^j$. For each pair of vertices $I^i, I^j$, we have

$$\begin{aligned}
\exists p, \ I^i \to I^0 \to I^j, \\
\exists p, \ I^j \to I^0 \to I^i.
\end{aligned} \tag{19}$$

From Eq.(19), $C$ is strongly connected which conflicts the definition of strongly connected components that each strongly connected component is maximal. Therefore, we prove that $C^i \cap C^j = \emptyset$. $\quad\square$

Theorem 2 states that the collaboration coalitions identified iteratively leads to a *collaboration equilibrium*. To prove the Corollary 1, we will demostrate that the collaboration coalitions identified iteratively are all strongly connected components of $BG(\boldsymbol{I})$ when Assumption 1 holds.

*Proof.* Suppose the coalitions strategy $S = \{C^0, C^1, ...C^k\}$ is determined iteratively and $C^i$ is identified in the i-th iteration. From Theorem 1, $C^0$ is one of the strongly connected components of $BG(\boldsymbol{I})$. $C^1$ is a stable coalition which is one of the strongly connected componts of $BG(\boldsymbol{I}\backslash C^0)$. When Assumption 1 holds, $BG(\boldsymbol{I}\backslash C^0)$ is the subgraph of $BG(\boldsymbol{I})$ and any edge in $BG(\boldsymbol{I}\backslash C^0)$ is also the edge in $BG(\boldsymbol{I})$ and vice versa. Therefore, $BG(C^1)$ is strongly connected in $BG(\boldsymbol{I})$. Suppose $BG(C^1)$ is not a strongly connected component of $BG(\boldsymbol{I})$ and there exists an additional client $I^i$ which can be added in $C^1$ without breaking the strongly connected property. From Lemma 2, $I^i$ cannot be from another strongly connected component, so we have

$$I^i \notin C^0. \tag{20}$$

If $I^i$ is from $\boldsymbol{I}\backslash(C^0 \cup C^1)$ and can be added into $C^1$ without breaking the strongly connected property, it conflicts the definition that $C^1$ is a strongly connected component of $\boldsymbol{I}\backslash C^0$. So we have

$$I^i \notin \boldsymbol{I}\backslash(C^0 \cup C^1). \tag{21}$$

From Eq.(20) and Eq.(21), there is no additional $I^i$ which can be added in $C^1$ without breaking the strongly connected property. So $C^1$ is a strongly connected component of $BG(\boldsymbol{I})$. For the same analysis, we can prove that $\forall C^i \in S$, $C^i$ is a strongly connected component of $BG(\boldsymbol{I})$ when Assumption 1 holds. Therefore, we prove that the strongly connected components of $BG(\boldsymbol{I})$ are the stable coalitions $\{C^0, C^1, ...C^k\}$ identified iteratively by Algorithm 1 and these strongly connected components lead to a CE. $\quad\square$

## A.4 Proof of Proposition 1

*Proof.* For any hypothesis $h'^* \in P(C')$, $h'^*$ satisfies

$$\nexists h' \in \mathcal{H}, \ \text{s.t.} \ \forall i \in C' : l_i\left(h'\right) \le l_i(h'^*) \ \text{and} \ \exists j : l_j\left(h'\right) < l_j(h'^*). \tag{22}$$

1) If $h^* \in P(C)$ which satisfies

$$\nexists h' \in \mathcal{H}, \ \text{s.t.} \ \forall i \in C : l_i\left(h'\right) \le l_i(h) \ \text{and} \ \exists j : l_j\left(h'\right) < l_j(h), \tag{23}$$

then we let $h^* = h'^*$ in Eq.(8) so Eq.(8) holds;

2) if $h'^* \notin P(C)$, there exists $h' \in P(C)$ which satisfies

$$\forall i \in C : l_i\left(h'\right) \le l_i(h'^*) \text{ and } \exists j : l_j\left(h'\right) < l_j(h'^*), \tag{24}$$

combining Eq.(22), we have

$$\forall i \in C' : l_i\left(h'\right) = l_i(h'^*) \text{ and } \exists j \in C \backslash C' : l_j\left(h'\right) < l_j(h'^*). \tag{25}$$

then we let $h^* = h'$ in Eq.(8) so Eq.(8) holds.

$\square$

## B  IMPLEMENTATION

In this section, we will introduce how to learn the Pareto Front of all objectives corresponding to all clients. Then, we will describe how to determine the optimal collaborator set for each client.

### B.1  LEARNING PARETO FRONT

Multi-objective optimization (MOO) problems have a set of optimal solutions, and these optimal solutions form the Pareto front, where each point on the front represents a different trade-off among possibly conflicting objectives. We construct a personalized model for each client which we call target network and it has the same architecture as the baselines. To determine the parameters of the target network of each client, in our experiments, we learn the entire Pareto Front simultaneously using a hypernetwork which receives as input a vector $d$ and returns all parameters of the target network. The input vector is N-dimension $bmd$ in which each entry $d^i$ corresponds to the client $I^i$ and is sampled from the convex hull $\mathcal{D}$,

$$\mathcal{D} = \left\{ d, |\forall i, 1 \le i \le N, d^i \ge 0, \text{ and } \sum_{i=1}^{N} d^i = 1 \right\}. \tag{26}$$

Specifically, we construct the hypernetwork following in the architecture introduced in (Navon et al., 2020). By a n-layer ($n \le 3$) MLP network with the activation function ReLU, we inference the parameters of the target network. Then the obtained parameters will be assigned to the target network. We evaluate the performance of the parameters on the training set and the gradient information will be returned for updating the hypernetwork.

### B.2  IDENTIFYING THE OPTIMAL COLLABORATOR SET

From the statement above, we train a hypernetwork $HN$ for learning the whole Pareto front. $HN$ bridges the mapping from the vector $d$ to the corresponding model parameters of the target network. To obtain the optimal target network parameters that achieve the minimal value loss of the target objective on the validation set, we optimize the vector $d$ by gradient descent. Specifically, given an initial direction $d_0$, we firstly obtain the parameters of the target network using the learned hypernetwork $HN(d_0)$. Like training the hypernetwork, the obtained parameters will be assigned to the target network. Then we evaluate the performance of the generated parameters on the validation set and compute the gradient of the input direction $d_0$. Finally, the gradient information will be used for updating the vector $d$ iteratively until convergence.

$$\begin{aligned}
d_{i+1} &= d_i - \eta \cdot \nabla_{d_i} l^* \\
d_{i+1} &\leftarrow Clip(d_{i+1}) \\
d_{i+1} &\leftarrow Normalization(d_{i+1}),
\end{aligned} \tag{27}$$

where $\eta$ denotes the learning rate, $Clip(d_{i+1})$ means that we clip each values $d_{i+1}^j \in d_{i+1}$ to satisfy $\epsilon_0 \le d_{i+1}^j \le 1$ and $Normalization(d_{i+1})$ is as follows,

$$\forall j \ (1 \leq j \leq N), d'^{j}_{i+1} = \frac{d^{j}_{i+1}}{\sum_{j=1}^{N} d^{j}_{i+1}}. \tag{28}$$

Finally, we reach the optimal direction $d^*$ and its corresponding Pareto model $M^* = HN(d^*)$.

For each client, we determine the optimal collaborator set by the optimal direction $d^*$ and the loss value of all objectives $l = [l_1(M^*), l_2(M^*), ..., l_N(M^*)]$. From the Pareto Front Embedded property described in Proposition 1, an ideal direction $d^*$ is a sparse vector in which the indexes of the non-zero values in $d^*$ are the necessary collaborators for the target client. In our experiments, $d^*$ usually is not a sparse vector, but there are values in $d^*$ which are significantly small. Therefore, we set a threshold $\epsilon$ for $d^*$ to determine which clients are necessary/unnecessary.

---

**Algorithm 1:** Achieving collaboration equilibrium

---

**Input:** $N$ institutions $\boldsymbol{I} = \{I^i\}_{i=1}^N$ seeking collaborating with others
Set original client set $C \leftarrow \boldsymbol{I}$;
Set collaboration strategy $S \leftarrow \emptyset$;
**while** $C \neq \emptyset$ **do**
    **forall** *client $I^i \in C$* **do**
        | Determine the OCS of $I^i$ by SPO;
    **end**
    Construct the benefit graph $BG(C)$;
    Search for all strongly connected components $\{C^1, C^2, ...C^k\}$ of $BG(C)$ using **Tarjan** algorithm;
    **forall** *i = 1, 2, 3,... k* **do**
        **if** $C^i$ *is stable coalition* **then**
            | $C \leftarrow C \backslash C^i$ ;
            | $S \leftarrow S \cup \{C^i\}$;
        **end**
    **end**
**end**
**Output:** collaboration strategy $S$

---

When Assumption 1 holds, from Corollary 1 proposed in the main text, the framework for obtaining a collaboration strategy that leads to a CE is simplified as in Algorithm 2.

---

**Algorithm 2:** Achieving collaboration equilibrium under Assumption 1

---

**Input:** $N$ institutions $\boldsymbol{I} = \{I^i\}_{i=1}^N$ seeking collaborating with others
Set original client set $C \leftarrow \boldsymbol{I}$;
Set collaboration strategy $S \leftarrow \emptyset$;
**forall** *client $I^i \in C$* **do**
    | Determine the OCS of $I^i$ by SPO;
**end**
Construct the benefit graph $BG(C)$;
Search for all strongly connected components $S = \{C^1, C^2, ...C^k\}$ of $BG(C)$ using **Tarjan** algorithm;
**Output:** collaboration strategy $S$

---

## C   EXPERIMENTAL DETAILS

### C.1   SYNTHETIC DATA EXPERIMENTS

The synthetic data generation process is illustrated in the main text. In this experiment, we construct the hypernetwork using 1-layer hidden MLP for generating the target network parameters and the

target network is a 1-layer MLP. The experiments have two settings, 1) 2000 training samples and $\rho = 0.1$; 2) 20000 training samples and $\rho = 1.0$.

We plot the vector $d = [d^0, d^1, d^2, d^3, d^4, d^5]$ varying in each iteration as we learn a personalized model for $I^0$. When $n = 2000$, $\rho = 0.1$ and the target client is $I^0$, from Figure 6(a), $\{d^3, d^4, d^5\}$ decrease until reach the minimum and $\{d^0, d^1, d^2\}$ increase in each iteration. As $\rho \neq 0$, the data in $I^1$ ($I^2$) and $I^0$ does not satisfy i.i.d, so $d^0$ is slightly larger than $d^1$ and $d^2$. This means that the optimal personalized model of $I^0$ is on $PF(I^0, I^1, I^2)$ and the OCS of $I^0$ is $\{I^0, I^1, I^2\}$. When $n = 20000$, $\rho = 1.0$ and the target client is $I^0$, from Figure 6(b), $\{d^1, d^2, d^3, d^4, d^5\}$ decrease and reach the minimum and $d^0$ increases in each iteration. This means that the optimal personalized model of $I^0$ is on $PF(I^0)$ and the OCS of $I^0$ is $\{I^0\}$.

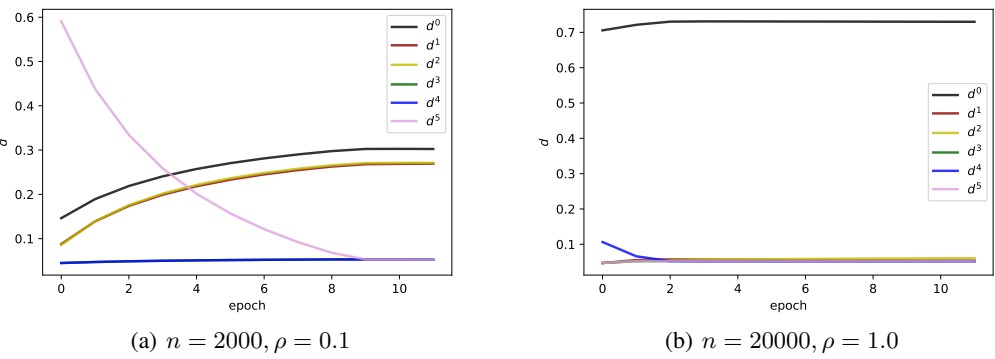

(a) $n = 2000, \rho = 0.1$  (b) $n = 20000, \rho = 1.0$

Figure 6: $d = [d^0, d^1, d^2, d^3, d^4, d^5]$ varies in each iteration where $d^i$ corresponds to the client $I^i$.

After identifying all coalitions, all clients will re-learn a personalized model by collaborating with other clients in its coalition.

## C.2 UCI ADULT DATA EXPERIMENTS

UCI Adult is a public dataset (Kohavi, 1996) which is split as training set and test set. We follow the dataset preprocessing procedure as in (Li et al., 2019; Mohri et al., 2019) and split the data into two clients PhD client and Non-PhD client. In this experiment, we construct the hypernetwork using a 1-layer hidden MLP. The target network is a Logistic Regression model as in (Li et al., 2019; Mohri et al., 2019). We split 83% training data for learning the PF and the remaining 17% training data for determining the optimal vector $d$. All hyperparameters are determined by the performance on the validation data set.

## C.3 CIFAR10 DATA EXPERIMENTS

CIFAR10 is a public dataset (LeCun et al., 1998) which contains 50000 images for training and 10000 images for testing. In our experiments, we following the setting in (McMahan et al., 2016). We split the data into 10 clients and simulate a non-i.i.d environment by randomly assigning two classes to each client among ten total classes. The training data of each client will be divided into training set (83%) and validation set(17%). We construct the hypernetwork using 3-layer hidden MLP for generating the parameters of the target network and the target network is constructed following the work (Shamsian et al., 2021). All baselines share the same target network structure for each client. To determine the hyperparameters (e.g., learning rate), we evaluate the performance of different hyperparameter combinations on the validation set and choose the hyperparameter combination which achieves the optimal accuracy.

## C.4 EICU DATA EXPERIMENTS

**eICU Data Asset** eICU dataset (Pollard et al., 2018) is a dataset for which permission is required. We followed the procedure on the website `https://eicu-crd.mit.edu` and got the approval

to the dataset. In this experiment, we follow the data preprocessing as in (Sheikhalishahi et al., 2019) and select 10 hospitals for collaboration as introduced in the main text. As all hospitals own relatively insufficient patient data samples ($100 \le n \le 1300$), dividing the data into training set and test set in different ways may lead to different CE. We construct the hypernetwork by a 1-layer MLP for training the Pareto Front of all objectives. The target network is an ANN model following the work in (Sheikhalishahi et al., 2019). We determine the hyperparameters by choosing the hyperparameter combination which achieves the optimal performance on the validation set. Full experiment results on eICU is in Table 5

Table 5: Full experimental results on eICU dataset.

| methods | ave | AUC | | | | | | | | | |
|---|---|---|---|---|---|---|---|---|---|---|---|
| | | $I^0$ | $I^1$ | $I^2$ | $I^3$ | $I^4$ | $I^5$ | $I^6$ | $I^7$ | $I^8$ | $I^9$ |
| Local | 66.44 | 66.89 | 85.03 | 61.83 | 68.83 | 82.31 | 59.65 | 67.78 | 40.00 | 61.90 | 70.00 |
| FedAve | 69.57 | 71.92 | 89.36 | 81.00 | 73.89 | 80.23 | 70.18 | 52.22 | 40.00 | 61.90 | 75.00 |
| Per-FedAve | 68.82 | 68.72 | 89.82 | 71.19 | 72.70 | 74.96 | 68.42 | 46.67 | 45.00 | 85.71 | 65.00 |
| FedPer | 72.53 | 72.59 | 87.74 | 80.01 | 56.37 | 86.56 | 94.74 | 73.33 | 15.00 | 85.71 | 73.33 |
| pFedMe | 70.68 | 76.21 | 89.31 | 76.30 | 68.17 | 87.50 | 82.46 | 84.44 | 10.00 | 52.38 | 80.00 |
| LG-FedAve | 71.07 | 74.70 | 88.80 | 76.37 | 66.45 | 86.40 | 82.46 | 83.33 | 15.00 | 57.14 | 70.00 |
| pFedHN | 67.57 | 57.29 | 76.15 | 84.67 | 52.01 | 66.95 | 45.61 | 57.78 | 70.00 | 95.24 | 70.00 |
| SPO (ours) | 75.90 | 76.35 | 91.80 | 80.28 | 70.52 | 86.93 | 82.46 | 71.11 | 40.00 | 76.19 | 83.30 |
| CE (ours) | 70.05 | 77.93 | 87.28 | 70.47 | 70.64 | 83.48 | 64.92 | 68.89 | 45.00 | 61.90 | 70.00 |

## C.5 TRAINING RESOURCES

We run our experiments on a local Linux server that has two physical CPU chips (Inter(R) Xeon(R) CPU E5-2640 v4 @ 2.40GHz) and 32 logical kernels. We use Pytorch framework to implement our model and train all models on GeForce RTX 2080 Ti GPUs.

## D DETAILED DISCUSSIONS ABOUT MULTI-DOMAIN ADAPTATION

Multi-domain adaptation refers to the learning scenario that the target domains have no labeled data and researchers transfer the knowledge by aligning the feature spaces between the source domains and the target domains (Mansour et al., 2008; Crammer et al., 2008), while the federated learning scenario in our work considers that all clients own labeled data and the feature spaces are shared. The differences between the methods in federated learning and multi-domain adaptation are as follows:

1. the core goal of the methods in multi-domain adaptation is to align the feature spaces between the source domains and the target domains (Wilson & Cook, 2020), which may be redundant for the problem we focus on because the feature space is shared across all domains;

2. in the paradigm of federated learning/multi-task learning, the data of the target clients are labeled and it allows researchers to learn an optimal model by supervised training while methods in multi-domain adaptation cannot;

3. federated learning requires that learning a model across multi-client cannot have access to the raw data of the local clients, while methods in multi-domain adaptation can.

## E EXTENDED DISCUSSIONS

### E.1 COLLABORATIVE EQUILIBRIUM

The collaboration equilibrium of a collaborated network is affected by 1) the selected model structure; 2) the data distribution of the clients. For 1), different model structures utilize the data in different ways. For example, non-linear models can capture the non-linear mapping relations in the data among different local clients while linear models cannot. For 2), the collaboration equilibrium is affected by the data distribution of the clients given the model structure. The data distribution of one client determines whether it can benefit another when learning together and this was verified

by the experiments. From the experimental results shown in Table 1, when the data distribution discrepancies increases ( from 0.1 to 1.0), $I^0$ and $I^1$ could not benefit $I^2$ anymore. Therefore, the collaboration equilibrium varies as shown in Figure 4(a).

### E.2 OPTIMALITY

**Identifying stable coalition to reach a CE**    Our method guarantees the optimality of achieving a CE given a benefit graph. In particular, the stable coalitions (or collaboration equilibrium) are determined uniquely given the benefit graph of the client set. Based upon the constructed benefit graph, our approach guarantees the optimality of the obtained stable coalitions theoretically as the identified stable coalitions strictly satisfy the two conditions in Definition 3 as proved in Sec A.2 in Appendix. Moreover, we point out that the strongly connected components of the benefit graph lead to a CE under Assumption 1 as proved in A.3;

**Constructing benefit graph by SPO**    Given the client set and the model structure, the true optimal collaborator sets (OCS) of each client may be defined uniquely. However, the true OCS of each client is actually agnostic without any prior assumption and we have to train an optimal model for each subset and then determine the OCS by observing which model performs best. Due to its exponential time complexity, we develop SPO to identify OCS by alternately optimizing the model and the collaborator set for each client. Roughly speaking, this procedure is similar to the Expectation-Maximization procedure performed in the mixture-of-experts (Masoudnia & Ebrahimpour, 2014) approaches, in which the model for each expert and the ensemble weights for each sample are also learned alternately. We verify its effectiveness on both synthetic and benchmark datasets. Specifically, from the experimental results shown in Figure 4, SPO finds the true OCS on synthetic, adult and cifar10.

### E.3 TIME COMPLEXITY

The analysis about the time complexity is as follows:

1. to find an OCS efficiently, we propose SPO to learn an optimal model by gradient descent and identify the OCS based upon the geometric location of the learned model on the Pareto Front according to Proposition 1, which avoids exhaustively trying;

2. to find the stable coalitions of a given client set, we propose to identify the OCS of all clients to construct the benefit graph firstly. Then we propose a graph-based method to recognize the stable coalitions which has O(V+E) time complexity as stated in Section 4.1;

3. to achieve a collaboration equilibrium, we propose to look for the stable coalitions and removes them iteratively. Combining (1) and (2), our method to reach CE has polynomial time complexity.

The run time of SPO is listed as in the following table. We also present the runtime of FedAve for comparison. As SPO learns the Pareto Front and optimizes the vector $d$ to search for the OCS by gradient descent, there is no significant additional time cost compared to baselines.

Table 6: Running Time on Different Datasets

| Dataset | FedAve | Ours |
|---------|--------|------|
| Adult | 14 min 36 s | 14 min 14 s |
| CIFAR10 | 1 h 24 min 52 s | 4 h 4 min 40 s |
| eICU | 21 min 30 s | 26 min 6 s |

### E.4 APPLICATIONS

One cannot know which one should collaborate with unless it knows the result of the collaboration. This requires that all clients agree to collaborate to obtain the results of the collaboration, before finalizing the collaboration equilibrium. The premise of achieving the collaboration equilibrium is that all clients agree to collaborate first to construct the benefit graph. In practice, we are inclined

to the view that this would be done by an impartial and authoritative third-party (e.g., the industry association) in the paradigm of federated learning. The approved third-party firstly determines the benefit graphs by learning the optimal models from multiple clients without direct access to the data of the local clients. Then the approved third-party derives the collaboration equilibrium based upon the benefit graphs and publishes the collaboration strategies of all clients.

As our framework quantifies the benefit and the contribution of each client in a collaborative network, on the one hand, this promotes a more equitable collaboration such as some big clients may no longer collaborate with small clients without any charge; on the other hand, it also leads to a more efficient collaboration as all clients will collaborate with the necessary collaborators rather than all participants.

### E.5 LIMITATIONS AND FUTURE WORK

In this work, we study the collaborative learning problem and propose the core goal that a collaboration strategy should lead to a *collaborative equilibrium* (CE). In theory, it is a combinatorial optimization problem with exponential complexity. There are plenty of works on addressing combinatorial optimization problems by reinforcement learning (Mazyavkina et al., 2021), though reinforcement learning methods are mostly time-consuming and hard to obtain stable results. Though we propose an efficient alternate-optimization method for CE in polynomial complexity, the optimality of the OCS (or the benefit graph) is still an unsolved problem that will be explored in our future work.

