# OpenReview forum: "Learning to Collaborate"
_ICLR.cc/2022/Conference — ICLR 2022 Submitted_

### Official Review · Reviewer_eyZ6 · 2021-10-29

**Correctness:** 4
**Technical Novelty And Significance:** 3
**Empirical Novelty And Significance:** 3
**Recommendation:** 8
**Confidence:** 2

**Main Review:**

### Strengths

The notion of collaboration equilibria appears natural, and the algorithm for computing such equilibria insightful.  Together with the SPO part, this provides a quite complete solution for personalized federated learning problems, and in particular, the equilibrium solutions take into consideration the incentives of self-interested agents, which is of potential practical importance.  The experiments appear to support the main claims of the paper.


### Weaknesses

Some parts of the overall approach lacks theoretical foundations (although they make sense).  The writing could be improved, e.g., by unifying terminologies and better connecting different parts of the paper.  Also, (this is not necessarily a weakness, but) I'm not familiar with the literature on federated learning, so I can't say much about the novelty of the method.


### Detailed comments

Problem setup, set of coalitions: since C^1, ... C^K partition I, isn't C^0 necessarily empty?

Axioms 1 and 2: these are conceptually quite similar to the notion of the core in cooperative game theory.  The authors may want to discuss the relation between the two.

Theorem 2: this claim is technically fine, but I think you still want to discuss why Algorithm 1 terminates, which isn't totally trivial (since very superficially it's possible that none of the components are stable so the algorithm gets stuck).  Indeed it does terminate, because after the SCC decomposition the new graph (over SCCs) is acyclic, which means it can be topologically ordered, and there exists a component which doesn't have incoming edges from other components.  That is a stable coalition that can be removed from C to make progress.

Definition 5: I'd say "Pareto-efficient (or Pareto-optimal) solution" instead of "Pareto solution"

Footnote 1: "(for) more information"

"Learning a best model" paragraph: what's a "model"?  What's a "Pareto model"?  From what I understand, a "model" is the same as a "solution" (which is a hypothesis), and a "Pareto model" is a solution or a hypothesis on the Pareto frontier.  Is that right?  In any case it would help to clarify these terms and unify them if they are in fact the same thing.

**Summary Of The Paper:**

The paper considers the problem of agents sharing training data to improve the accuracy of the model obtained on their own population.  The high-level goal is to find "stable" collaboration patterns where no agents want to deviate (e.g., stop sharing or form their own groups of sharing).  The authors first consider an abstract formulation where each agent has a utility function over all sets of other agents to collaborate with.  Assuming that one can find the most beneficial set of collaborators for each agent, the authors present an efficient algorithm to divide agents into groups that form a stable collaboration pattern (aka a collaboration equilibrium).  Since the problem of finding this optimal set of collaborators for each agent is nontrivial in general, the authors come back to the setting of sharing training data, where they propose a theoretically principled approach to build the benefit graph (which encodes most beneficial collaborators) by optimizing on the Pareto frontier of hypotheses / models of accuracy of all agents.  The authors then evaluate their approach on synthetic and real data, and compare that against several benchmark methods for personalized federated learning.  Empirical results suggest that the collaboration equilibrium behaves roughly as one would expect.  Also, the model found by optimizing on the Pareto frontier appears to achieve remarkable performance, often comparable to or better than the best benchmark.  The accuracy at collaboration equilibria is somewhat worse, which is natural given that there might be a "price of stability".

**Summary Of The Review:**

Overall I think this paper proposes a novel theoretically principled method for computing stable outcomes of data sharing or collaboration in general, which can also be applied to personalized federated learning.  Some of the theoretical results appear quite insightful.  The experiments are quite informative and support the main claims of the paper.  (Since I'm not familiar with the literature on federate learning, my evaluation regarding the novelty could be misled.)

---

> ### Author Response · Authors · 2021-11-10
> **Response to Reviewer eyZ6**
>
> We would like to thank the reviewer for the insightful and valuable feedback. Below are our response to the comments.
>
> To the comments in **Detailed comments**,
>
> * **Problem setup, set of coalitions: since C^1, ... C^K partition I, isn't C^0 necessarily empty?**
>
>     We would like to thank the reviewer for pointing out this typo error. It should be that $S = $ {$C^{0}, C^{1},..., C^{k}$} such that $\bigcup_{k=0}^{K} C^{k}=\boldsymbol{I}$ as you understand.
>
> * **Axioms 1 and 2: these are conceptually quite similar to the notion of the core in cooperative game theory. The authors may want to discuss the relation between the two.**
>
>     As the reviewer mentioned, axioms 1 and 2 are conceptually similar to the notion of the core in cooperative game theory. However, there are significant differences between the two. In cooperative game theory, the payoff is transferable among players in the coalition, and there is a predefined payoff function on all coalitions. In federated learning, the "payoff" is defined as the maximum achievable utility (MAU) on all players rather than coalitions. It is fixed and agnostic. Detailed demonstration please refer to **Response to Reviewer yALe**.
>
> * **Theorem 2: this claim is technically fine, but I think you still want to discuss why Algorithm 1 terminates, which isn't totally trivial...**
>
>     We would like to thank the reviewer for the kind suggestion. As you mentioned, the new graph (over SCCs) is acyclic and there exists a component without incoming edges from other components. We also give detailed proof by contradiction. Detailed illustration can be found in **Response to Reviewer Psa6 (part 1)**.
>
> * **Definition 5: I'd say "Pareto-efficient (or Pareto-optimal) solution" instead of "Pareto solution"** and **Footnote 1: "(for) more information,"**,
>
>     We would like to thank you for pointing out these mistakes. We will revise our paper carefully following your suggestions.
>
> * **"Learning a best model" paragraph: what's a "model"? What's a "Pareto model"? From what I understand, a "model" is the same as a "solution" (which is a hypothesis), and a "Pareto model" is a solution or a hypothesis on the Pareto frontier. Is that right? In any case, it would help to clarify these terms and unify them if they are in fact the same thing.**
>
>     We would like to thank you for your helpful advice. As you understand, a Pareto model is a solution or a hypothesis on the Pareto Front. We will clarify these terms and unify them to facilitate the reader to understand.
>
> * **(Since I'm not familiar with the literature on federate learning, my evaluation regarding the novelty could be misled.)**
>
>     We would like to explain that we consider a more practical federated learning scenario. Detailed illustration about the difference between our work and prior work in federated learning is in **Response to Reviewer yALe.**

---

### Official Review · Reviewer_Mxha · 2021-11-02

**Correctness:** 4
**Technical Novelty And Significance:** 2
**Empirical Novelty And Significance:** 2
**Recommendation:** 5
**Confidence:** 3

**Main Review:**

I think the paper solves an interesting and important problem and that the model introduced is interesting and detailed. My main issue is that it does not appear that the introduced model captures the problem. Here are some points:

1-The authors mention that privacy concerns are the main reason why a client would not share his data. But I don't see how the privacy cost is being modelled in the problem. For example, the number of individuals the client has to share his data with.

2-Following point 1: I don't think the paper mentions this, but is it not possible for clients to share portions of their datasets instead, perhaps the portions that lead to positive transfer. This would further complicate things and also make it more meaningful to consider the amount of lost privacy.

3-Similarly, the benefit graph does not include weights. But it seems plausible that a client would only benefit little from others datasets.

4-Assumption 1: is there a motivation behind this assumption? do we expect real instances to satisfy it or it is only introduced to make the algorithm less time consuming? Also given the benefit graph can we verify that the assumption holds?

5-Section 4.2 discusses how the benefit graph is obtained. It is not clear to me however, that the Pareto solution gives the actual benefit graph. I realize that the exhaustive search for the benefit graph would be infeasible. The issue is if this is an approximation of the benefit graph, then the error should be characterized and the algorithms and theorems in section 4.1 should be modified to reflect that they operate on a noisy estimate of the benefit graph.

6-Unrelated to the model, but the number of clients in the experiments seems to be small (at most 9) even on the synthetic dataset.

**Summary Of The Paper:**

Given a network of clients each with his own private dataset, the locally learned model can be improved by obtaining data from other clients (assuming no domain mismatch). The paper models this problem by introducing the benefit graph which models how each client may benefit another. The concept of collaboration equilibrium is introduced and optimization algorithms to solve it are given. Some theoretical guarantees are shown along with experimental results.

**Summary Of The Review:**

The introduced model does not seem to be complicated enough to capture the real details of the problem. Further, the algorithms and guarantees introduced do not seem to be rigorously justified.

---

> ### Author Response · Authors · 2021-11-11
> **Response to Reviewer Mxha (part 1)**
>
> We would like to thank the reviewer for the valuable feedback. Below are our response to the comments.
>
> * **1-The authors mention that privacy concerns are the main reason why a client would not share his data. But I don't see how the privacy cost is being modeled in the problem. For example, the number of individuals the client has to share his data with.**
>
>      would like to explain that the privacy cost is considered in the manner of training models, as prior works in federated learning do [1]:
>
>     1). our work focus on learning a local model for each client in federated learning setting. Therefore, all clients will not share any data with others. Only gradient information is allowed to be shared in federated learning [1];
>
>     2). we propose a gradient-based framework in Sec 4.2. Different from traditional federated learning, we propose to share the gradient information only in a subset that is called **stable coalition**, which further preserves privacy compared with sharing the gradient information with all clients in federated learning.
>
> * **2-Following point 1: I don't think the paper mentions this, but is it not possible for clients to share portions of their datasets instead, perhaps the portions that lead to positive transfer. This would further complicate things and also make it more meaningful to consider the amount of lost privacy.**
>
>     As we stated above, for privacy-preserving, each client is not allowed to share any data with another client. We agree with the reviewer that the portions of the datasets may lead to positive transfer, and it is forbidden in federated learning. As you mentioned, considering the trade-off between the portions of data sharing and privacy-preserving may be a more meaningful research direction and this is worthy of further exploration.
>
> * **3-Similarly, the benefit graph does not include weights. But it seems plausible that a client would only benefit little from other datasets.**
>
>     1). As you mentioned, the benefit graph does not include weights. This is because it is hard to exactly quantify the benefit of one client to another, especially when the two clients are in different client sets. Therefore, we propose the definition **optimal collaborator set** (OCS) to represent the most helpful client set for a given target client.
>
>     2). We agree with the reviewer that a client would only benefit little from other datasets. In our framework, a client will collaborate with others even it benefits little. To address this problem, we can set a reasonable bound $\epsilon$, and only when the benefit exceeds $\epsilon$ can the collaboration happen.
>
> * **4-Assumption 1: is there a motivation behind this assumption? do we expect real instances to satisfy it or it is only introduced to make the algorithm less time consuming? Also given the benefit graph can we verify that the assumption holds?**
>
>     We would like to explain that this assumption is not just for simplifying our algorithm.
>
>     1). The motivation behind this assumption is that for two clients $I^{i}$ and $I^{j}$, the impact of $I^{j}$ on $I^{i}$ is not reversed by other clients. In particular, they benefit each other means that the data distribution of the two clients share some common patterns which can be captured and utilized by the model. Meanwhile, these shared patterns will not be affected by other clients.
>
>     2). Given the benefit graph, we can verify whether the assumption holds by comparing the full benefit graph and the benefit graph of each subset. However, the true benefit graph is agnostic and can only be approached by exhaustively trying all subsets. From our experiments, we found that a helpful client will not become a harmful client whether other clients join the collaboration or not. This verifies the rationality of Assumption 1;
>
>
>
> [1] Brendan McMahan, Eider Moore, Daniel Ramage, Seth Hampson, and Blaise Aguera y Arcas. Communication-efficient learning of deep networks from decentralized data. In Artificial Intelligence and Statistics, pp. 1273–1282. PMLR, 2017.

---

> ### Author Response · Authors · 2021-11-11
> **Response to Reviewer Mxha (part 2)**
>
> * **5-Section 4.2 discusses how the benefit graph is obtained. It is not clear to me however, that the Pareto solution gives the actual benefit graph. I realize that the exhaustive search for the benefit graph would be infeasible. The issue is if this is an approximation of the benefit graph, then the error should be characterized and the algorithms and theorems in section 4.1 should be modified to reflect that they operate on a noisy estimate of the benefit graph.**
>
>     We would like to explain that 1). **the optimality of the obtained benefit graph is reflected in the optimality of the model learned based on the benefit graph.** In particular, if a model learned based on the benefit graph achieves a SOTA performance, this benefit graph is practical and meaningful. In our experiments, we first obtain the benefit graph using the proposed algorithm in Sec 4.2, then we re-learn an optimal model with the learned OCS encoded in the obtained benefit graph. From the experimental results, our method achieves SOTA performances on both synthetic and real-world datasets. This demonstrates that the obtained benefit graph is reasonable;
>
>     2). as you mentioned that the exhaustive search for the benefit graph would be infeasible. However, deriving a theoretical error may be impossible without an exhaustive search. Because without any priori assumption, the true benefit graph will be agnostic forever, unless we do an exhaustive search. Therefore, we obtained the true benefit graph by an exhaustive try in our synthetic data sets. For example, in the first experiment, we learned $2^{6}-1$ models for each client to obtain the true benefit graph. The results in Figure 4 show that our method in Sec 4.2 captures the true benefit graph;
>
>     3). in reality, obtaining the true benefit graph is intractable because of its exponential time complexity. To tackle this problem, we propose to evaluate the effectiveness of the benefit graph by observing the performance of the learned model based on the benefit graph. Experimental results in the main text verify that optimizing the weights of all clients can approach the true benefit graph.
>
>
> * **6-Unrelated to the model, but the number of clients in the experiments seems to be small (at most 9) even on the synthetic dataset.**
>
>     We would like to explain that, to verify that our method can approach the true benefit graph, we need to exhaustively try all subsets to construct the true benefit graph, although it has the exponential time complexity. For example, in the synthetic experiment on CIFAR10, we tried $2^{10} - 1$ different subsets to determine the true optimal collaborator set (OCS) for each client. The results in Figure 4 show that only when all clients join the collaboration will the models of all clients have the best performance. A larger client set, for example, 50 clients, requires us to try $50 * (2^{50} -1)$ models to obtain the true benefit graph, which may be unrealistic. Therefore, the number of clients is at most 10 in our synthetic experiments.

---

> ### Author Response · Authors · 2021-11-23
> **Need Further Clarification?**
>
> Thanks very much for your constructive comments on our work. We have tried our best to address the concerns. Is there any unclear point so that we should/could further clarify?

---

### Official Review · Reviewer_yALe · 2021-11-08

**Correctness:** 4
**Technical Novelty And Significance:** 2
**Empirical Novelty And Significance:** 2
**Recommendation:** 3
**Confidence:** 2

**Main Review:**

The main aspect of the proposed solution in the paper is the notion of maximum achievable utility (MAU) and optimal collaborator set (OCS) which each agent must derive.  These are well-studied principles in cooperative game theory and concepts like 'core' and 'stable coalition' are analogous to the definitions used in this work.  Similarly, the approach of using a 'benefit graph' to compute coalitions satisfying the proposed axioms also has an equivalent approaches.  See , for example, literature on "stable group formation" (Y. Bachrach, V. Syrgkanis, and M. Vojnovic, 2013; Arkin et. al., Geometric Stable Roommates, 2009; Aauman and Dreze, Cooperative games with coalition structures, 1974; Hart and Kurz, Stable Coalition Structures, 1982).

Further, the authors have not motivated clearly why this should be a learning problem and why it cannot be solved by using other techniques (example, in the literature mentioned above).  In its current form, I think the paper needs more work to distinguish itself from prior art and how it can use the federated learning framework to 'learn' the OCS instead of computing it directly.

**Summary Of The Paper:**

The authors propose a model of collaboration to improve outcomes for any participating agent.   In their setting, the authors assume that every agent does not benefit from always collaborating with all other agents because of heterogeneity of the underlying data distributions.

**Summary Of The Review:**

There is a lot of related work in the cooperative game theory and federated learning literature.  I think the authors need to do a much better comparison with the existing literature and propose why their work is different from it and carry out clear comparisons (in their experiments) to show the efficacy of their approach.

---

> ### Author Response · Authors · 2021-11-10
> **Response to Reviewer yALe**
>
> We would like to thank the reviewer for the valuable feedbacks. Below are our response to the comments.
>
> * to the comment about **the difference between our work and cooperative game theory**,
>
>     1). **different definition:** cooperative game theory[1, 2, 3] requires a predefined payoff function $V(C)$ on all subsets $C \in S$, so that it can obtain the payoff of each subset $V(C)$ directly. In federated learning, the payoff is the maximum achievable utility ($U(I^{i}, C)$) of each client in $I^{i} \in C$, which is not the function of the subset $C$. What's more, $U(I^{i}, C)$ is agnostic and fixed given the client $I^{i}$, so we need to learn an optimal model to approximate $U(I^{i}, C)$. However, in cooperative game theory, the payoff is the function of the subset $C$. It is transferable among players in the subset, so it can allocate the payoff among players arbitrarily, as long as the sum of all the payoff is $V(C)$;
>
>     2). **different goal:** cooperative game theory [1, 2, 3] aims to find a reasonable allocation of the payoff for all players given the function $V$, which is called a "core". However, in federated learning, the payoff of each client (MAU) is not transferable, because it is the achievable highest performance of the learned model. Our work aims to learn the best model for each client, which may not be related to the payoff allocation;
>
>     3). **different approach:** in federated learning, there is no such predefined payoff function as in cooperative game theory. More importantly, the payoff of each client is fixed rather than transferable. Therefore, prior methods in cooperative game theory [1, 2, 3] for deriving a "stable group formation" cannot be used in our problem. Nevertheless, suppose we define the MAU of each player as the maximum achievable payoff in cooperative game theory. The iterative graph-based method we proposed can be directly used for identifying the coalition structure in cooperative game theory. **From the above analysis, our research extends prior work in cooperative game theory when the payoff of each coalition is not transferable among players.**
>
>     Since our work focuses on improving the performance of the local model in federated learning, we only present the most related work (federated learning, multi-task learning) in our main text. Following your suggestions, we will discuss the difference between our work and cooperative game theory in our final version.
>
>     **Additional information about this can be found in Official Review of Paper200 by Reviewer eyZ6.**
>
> * to the comment about **why the OCS needs to be learned**,
>
>     as we stated above, we define the payoff of each client in $C$ as the achievable highest performance (MAU), which is not predefined or even agnostic. We can only try to learn the best model to approximate the payoff. Since there is no such payoff function defined over the subset and transferability of the payoff, we cannot directly compute the OCS as cooperative game theory does.
>
>     The OCS is the necessary collaborator for achieving the highest accuracy. To determine the OCS, exhaustively choosing every subset to learn models can have exponential time complexity. To identify the OCS effectively and efficiently, we propose to learn the best model and optimize the collaborator set on the full Pareto Front in Sec 4.2.
>
> * to the comment about **the difference between our work and prior work in federated learning**,
>
>     1). **different setting**, our work considers a more practical scenario. Prior work in federated learning assumes all clients are willing to participate in collaboration, even if they may not gain benefits or even suffer losses. However, we consider that a client will refuse to join the collaboration to provide "free lunch" for other clients unless it benefits from the collaboration.
>
>     2). **different goal**, prior work in federated learning only cares about improving the utility for each client. It does not care about which clients are helpful/harmful. However, our work needs to identify which clients are necessary for achieving the maximum utility to achieve a collaboration equilibrium. Therefore, we propose to identify the OCS in addition to learning the best model.
>
> [1] Geometric Stable Roommates. Esther M. Arkin, Alon Efrat, Joseph S. B. Mitchell, Valentin Polishchuk, 2007.
>
> [2] Cooperative games with coalition structures.  R. J. Aumann, Jerusalem, and J. H. Dreze, Louvain, 1974.
>
> [3] Stable Coalition Structures. S. Hart, Tel Aviv, and M. Kurz, 1984.

---

> ### Author Response · Authors · 2021-11-23
> **Need Further Clarification?**
>
> Thanks very much for your constructive comments on our work. We have tried our best to address the concerns. Is there any unclear point so that we should/could further clarify?

---

### Official Review · Reviewer_Psa6 · 2021-11-09

**Correctness:** 3
**Technical Novelty And Significance:** 2
**Empirical Novelty And Significance:** 3
**Recommendation:** 5
**Confidence:** 4

**Main Review:**

Strengths:
- The authors consider an interesting problem as negative transfer is a real concern in federated learning and building a local model to avoid this phenomenon seems like a good direction.
- I like the notion of collaboration equilibrium where a group of agents share their data only within clients in a given coaltion.

Weaknesses:
- My main concern about the proposed iterative algorithm is that it works only when assumption 1 holds and I believe this is not a realistic assumption in practice.
- Although the definition is proposed for a general utility model, the authors assume a particular utility model in subsection 4.2. I have some concerns about this choice which I mention below.
- I don't think the proposed definition properly captures the negative transfer phenomenon we observe in federated learning. For example, negative transfer also happens when a client shares too much data from another client as opposed to a small amount of data.
- This paper talks about collaboration equilibrium but such notions of collaboration have been studied in cooperative game theory, probably in different context. I was surprised to see that the related work section doesn't have any mention of related work from game theory literature.

Questions for the authors:
- Regarding the example shown in figure 1, it is not clear that you will always be able to find an initial coalition from the graph. It was later clear that under assumption 1, this problem boils down to finding a strongly connected component but this was not clear in the introduction.
In particular, I wonder how the iterative algorithm would start when assumption 1 fails.

- Although the definition of optimal collaborator set is intuitive I am not sure how this captures the negative transfer phenomenon the authors talked about in the introduction. Negative transfer should imply that whenever I add a (harmful) client, the utility of the local model always drops. I don't think definition (1b) captures it exactly.

- How realistic is assumption 1 in practice? I understand that this assumption simplifies the iterative algorithm as you don't have to recompute the benefit graph after eliminating each coalition. But if this assumption fails, recomputing the benefit graph shouldn't be computationally hard.

- In section 4.2, I didn't understand the particular modeling assumptions about the clients' utility functions. In particular, it seems that the goal is to find a hypothesis that is at the pareto frontier. But why care about such a universal function if each agent wants to build their own local model? Additionally, how do the agents determine their weights?

**Summary Of The Paper:**

This paper considers the problem of federated learning where different clients/entities share their datasets with other clients in order to obtain a model that performs best for the local loss functions. Standard federated learning setting usually share all datasets together to obtain a central model and might not be the best choice for each client separately. The authors propose a notion of collaboration equilibrium where each client shares data with a subset of all the available clients. In particular, a coalition of agents belong to a collaboration equilibrium if each agent gets maximum benefit by collaborating with all the agents in the subset, and the subset is maximal.

In order to obtain such a collaboration equilibrium the authors propose an iterative algorithm which under certain assumptions, is equivalent to finding strongly connected components in the graph. Finding the collaboration equilibrium requires identifying the benefit graph among the client. Under a very specific model of agents' utility functions, a pareto optimization based algorithm is proposed to find the graph. Finally, though experiments, the authors show that finding collaboration equilibrium does provide improvement compared to standard federated learning based methods.


**Summary Of The Review:**

I thought the authors consider an interesting problem in this paper. Negative transfer in federated learning is a challenging problem and if you can avoid this issue by building a local model, that would be great! However, there seems to be several issues with the proposed definition. Moreover, the methods work under certain assumptions and I am not sure the assumptions are realistic.

---

> ### Author Response · Authors · 2021-11-10
> **Response to Reviewer Psa6 (part 1)**
>
> We would like to thank the reviewer for the valuable feedback. Below are our response to the comments.
>
> * to the comment in **Weakness**,
>
>     1). for our proposed algorithm when Assumption 1 does not hold, we clarify it carefully in the next key point;
>
>     2). for the concerns in subsection 4.2, our detailed clarification is in the following;
>
>     3). for the concerns about the negative transfer phenomenon, we agree with the reviewer that negative transfer can happen when using too much data from another client. However, federated learning does not allow sharing any part of data due to privacy concerns [1]. In other words, **collaboration in federated learning is binary (providing the statistical gradient information of all data, or does not providing anything), as stated in Official Review of Paper200 by Reviewer Mxha.** Under the constraint that only the overall gradient information can be shared, we capture negative transfer by defining maximal achievable utility. It learns the best model with the highest performance for each client. During training, it will automatically determine the optimal weights of other clients to avoid such negative transfer. More discussion about it is in the following.
>
>     4). for the concerns about cooperative game theory, we would like to explain that due to space limitations, we only discuss the most related work in our main text, and we will follow your suggestions to revise our paper. For a detailed discussion about the difference between cooperative game theory and our work, please refer to **Response to Reviewer yALe.**
>
>
> * **Question:** **"Regarding the example shown in figure 1, it is not clear that you will always be able to find an initial coalition from the graph. It was later clear that under assumption 1, this problem boils down to finding a strongly connected component but this was not clear in the introduction. In particular, I wonder how the iterative algorithm would start when assumption 1 fails."**,
>
>     **Answer:** Our iterative algorithm is not affected by Assumption 1. Assumption 1 just guarantees the re-build benefit graph remains the same as the original graph. Whether Assumption 1 holds or not, there always exists one (or more) stable coalition given an arbitrary benefit graph. We can briefly prove it as follows.
>
>     1). From our proposed method, firstly, we search for all strongly connected components (SCC) { $C^{1}, C^{2},...,C^{k}$}. From Definition 4, these SCCs consist of a partition of all clients. For convenience, we group each SCC as a point in the benefit graph.
>
>     * (1). Suppose all SCCs are not stable coalitions, this means that each SCC $C^{i}$ cannot achieve its optimal performance without the collaboration with clients in other SCCs;
>
>     * (2). from (1), for each $C^{i}$, there exists at least an edge pointing to $C^{i}$ (since it needs other SCCs' help);
>
>     * (3). from graph theory, if for each node $C^{i}$, there is an edge pointing to $C^{i}$, then there exists a loop in this graph.
>
>     * (4). from (3), this loop means there is a larger SCC in the benefit graph, which contradicts the definition of SCC that SCC is maximal.
>
>     * So we prove that there always exists a $C^{i}$ that no edge points to, and such a SCC is a stable coalition.
>
>     In fact, the graph of SCCs is a directed acyclic graph and there always exists a node (SCC) that no edge points to (which is called "head node"). For example in Figure 1 (a), the graph of SCCs is {$I^{1}, I^{2}, I^{3} $} $\rightarrow$  {$I^{4}$}  $\rightarrow$ { $I^{5}, I^{6}$}, in which {$I^{1}, I^{2}, I^{3}$} is a stable coalition that no edge points to.
>
>     2). Assumption 1 cannot let a SCC be a stable coalition. It only guarantees that the re-build benefit graph of the remaining clients remains unchanged after we remove the stable coalitions. Assumption 1 does not boil down our problem to finding all SCCs. Because we prove that under Assumption 1, if a SCC in the original benefit graph is not a stable coalition, it will always be identified as a stable coalition along with more and more stable coalitions being removed in each iteration.
>
>     For example, {$I^{5}, I^{6} $} and {$I^{4} $} are SCCs but not stable coalitions in the original benefit graph in Figure 1. However, after the stable coalition {$I^{1}, I^{2}, I^{3}$} are removed, {$I^{4}$} is a stable coalition when the benefit graph of {$I^{4}, I^{5}, I^{6}$} remains unchanged. In a similar manner, {$I^{4}$} will be removed and {$I^{5}, I^{6}$} becomes a stable coalition when the benefit graph of {$I^{5}, I^{6}$} remains.
>
> [1] Brendan McMahan, Eider Moore, Daniel Ramage, Seth Hampson, and Blaise Aguera y Arcas. Communication-efficient learning of deep networks from decentralized data. In Artificial Intelligence and Statistics, pp. 1273–1282. PMLR, 2017.

---

> ### Author Response · Authors · 2021-11-10
> **Response to Reviewer Psa6 (part 2)**
>
> * **Question: "Although the definition of optimal collaborator set is intuitive I am not sure how this captures the negative transfer phenomenon the authors talked about in the introduction. Negative transfer should imply that whenever I add a (harmful) client, the utility of the local model always drops. I don't think definition (1b) captures it exactly."**,
>
>     **Answer:** 1). Definition 1(b) emphasizes that each client in the optimal collaborator set (OCS) is necessary. Because the utility will drop if any client is removed from OCS.
>
>     2). Definition 1(a) captures the negative transfer exactly by requiring a maximal utility in federated learning. As you mentioned, whenever I add a (harmful) client, the utility of the local model always drops, if we truly use this harmful client for training. However, if we try to learn the best model after we add a harmful client. The best model will not use the gradient information from this harmful client to keep its superiority. Meanwhile, our proposed framework in Sec 4.2 allows assigning a zero weight to this harmful client. The utility will not be a maximum if negative transfer happens. Therefore, our definition captures negative transfer by maximizing the utility of the local model, in federated learning setting.
>
> * **Question: "How realistic is assumption 1 in practice? I understand that this assumption simplifies the iterative algorithm as you don't have to recompute the benefit graph after eliminating each coalition. But if this assumption fails, recomputing the benefit graph shouldn't be computationally hard."**,
>
>     **Answer:** 1). As we mentioned in the main text, Assumption 1 implies that for two clients $I^{i}$ and $I^{j}$, the impact of $I^{j}$ on $I^{i}$ will not reverse by other clients. In particular, they benefit each other means that the data distribution of the two clients share some common patterns which can be captured and utilized by the model. Meanwhile, these shared patterns between the two data distributions will not be affected by other clients. From our experiments, we found that a helpful client will not become a harmful client whether other clients join the collaboration or not. This verifies the rationality of Assumption 1.
>
>     2). As you mentioned, recomputing the benefit graph may not be computationally hard, and our proposed iterative algorithm does not require Assumption 1 to hold. All experiments are conducted without Assumption 1. For example, we re-build the benefit graph on eICU data set after eliminating the coalition {$I^{0}, I^{3}$} and {$I^{9}$}, the experimental results show that the benefit graph on {$I^{1}, I^{2}, I^{4}, I^{5}, I^{6}, I^{7}, I^{8}$} is still a whole SCC as presented in Figure 5.
>
> * **Question: "In section 4.2, I didn't understand the particular modeling assumptions about the clients' utility functions. In particular, it seems that the goal is to find a hypothesis that is at the Pareto frontier. But why care about such a universal function if each agent wants to build their local model? Additionally, how do the agents determine their weights?"**
>
>     **Answer:** 1). **"why care about such a universal function,"** In federated learning, we have no access to the raw data of other clients due to privacy concerns. The most classic and universal way to use the knowledge from other clients is to use the gradient information [1] of the objectives of other clients. Intuitively, caring about the utility of other clients means the learned local model borrows the knowledge in the data from other clients. A Pareto model which cares about all objectives means better utilization of the data from all clients. Because of the insufficient data in local clients, a learned model with high accuracy on its own data can have high variance. Therefore, higher performance on other clients may indicate a better generalization.
>
>     2). **"how do the agents determine their weights,"** We propose to use gradient descent to optimize the weights of all clients. In summary, we first learn the full Pareto Front using a Hypernetwork, which outputs a Pareto model according to the input weight vector. Then we optimize the weights of all clients by gradient descent to maximize the performance of the output Pareto model on the validation set. Detailed information about the implementation is in Appendix B.
>
>  [1] Brendan McMahan, Eider Moore, Daniel Ramage, Seth Hampson, and Blaise Aguera y Arcas. Communication-efficient learning of deep networks from decentralized data. In Artificial Intelligence and Statistics, pp. 1273–1282. PMLR, 2017.

---

> ### Author Response · Authors · 2021-11-10
> **Response to Reviewer Psa6 (part 3)**
>
> * for the weakness **My main concern about the proposed iterative algorithm is that it works only when assumption 1 holds and I believe this is not a realistic assumption in practice.** and  the first question **Regarding the example shown in figure 1, it is not clear that you will always be able to find an initial coalition from the graph. It was later clear that under assumption 1, this problem boils down to finding a strongly connected component but this was not clear in the introduction. In particular, I wonder how the iterative algorithm would start when assumption 1 fails.**
>
>     In addition to a detailed proof by contradiction in Response to Reviewer Psa6 (part 1), Reviewer eyZ6 also gives a more creative explanation: **Theorem 2: this claim is technically fine, but I think you still want to discuss why Algorithm 1 terminates, which isn't totally trivial (since very superficially it's possible that none of the components are stable so the algorithm gets stuck). Indeed it does terminate, because after the SCC decomposition the new graph (over SCCs) is acyclic, which means it can be topologically ordered, and there exists a component which doesn't have incoming edges from other components. That is a stable coalition that can be removed from C to make progress.**

---

> ### Author Response · Authors · 2021-11-23
> **Need Further Clarification?**
>
> Thanks very much for your constructive comments on our work. We have tried our best to address the concerns. Is there any unclear point so that we should/could further clarify?

---

### Author Response · Authors · 2021-11-11
**Thanks for the four anonymous reviewers**

We are very grateful to the reviewers for their insightful feedback and suggestions. Our responses to the main concerns are given to each reviewer separately. Moreover, we want to know whether there are **unresolved or new concerns** we need to clarify and we are very pleased to discuss them further, if any.

Thank you again!

---

### Decision · Program_Chairs · 2022-01-20

**Decision:**

Reject

**Comment:**

The paper proposes a model of agent collaboration to improve outcomes for any participating agent in a setting where every agent does not always benefit from collaborating with all other agents. The reviewers did find some of the theoretical results interesting, however, in its current (revised) form, they still argued during the discussion post-rebuttal that: (i) the game theoretic formulation of this problem is not entirely new and has been studied in various forms before and (ii) the particular application of the results to federated learning comes after making various (questionable) assumptions. I would encourage the authors to take into account (i-ii) for preparing a revised version of their paper and resubmit to another conference.